# Maximal Brain Damage Without Data or Optimization: Disrupting Neural Networks via Sign-Bit Flips

**Ido Galil** *
*idogalil.ig@gmail.com, igalil@nvidia.com*
*NVIDIA*

**Moshe Kimhi** *
*moshekimhi91@gmail.com, moshekimhi@cs.technion.ac.il*
*Technion*

**Ran El-Yaniv**
*rani@cs.technion.ac.il, relyaniv@nvidia.com*
*Technion, NVIDIA*

**Reviewed on OpenReview:** *https://openreview.net/forum?id=kN1s53X3zl*

## Abstract

Deep Neural Networks (DNNs) can be catastrophically disrupted by flipping only a handful of parameter bits. We introduce Deep Neural Lesion (DNL), a data-free and optimization-free method that locates critical parameters, and an enhanced single-pass variant, 1P-DNL, that refines this selection with one forward and backward pass on random inputs. We show that this vulnerability spans multiple domains, including image classification, object detection, instance segmentation, and reasoning large language models. In image classification, flipping just two sign bits in ResNet-50 on ImageNet reduces accuracy by 99.8%. In object detection and instance segmentation, one or two sign flips in the backbone collapse COCO detection and mask AP for Mask R-CNN and YOLOv8-seg models. In language modeling, two sign flips into different experts reduce Qwen3-30B-A3B-Thinking from 78% to 0% accuracy. We also show that selectively protecting a small fraction of vulnerable sign bits provides a practical defense against such attacks.

## 1 Introduction

Deep neural networks (DNNs) now underpin a wide range of systems, from vision models to reasoning large language models. Their deployment in safety-critical and economically important settings raises an immediate security question: how much access and computation does an attacker need in order to induce severe failure? In this work we show that, once an attacker can write to stored parameters, the answer can be disturbingly small.

We expose a cross-domain vulnerability in DNNs that allows severe disruption by flipping only a few carefully chosen parameter bits. We systematically analyze and identify the parameters most susceptible to sign flips, which we term "critical parameters." By flipping a tiny number of these critical sign bits, an attacker can catastrophically damage deep neural network models across various domains, including classification, detection, and segmentation systems, as well as large language models. Crucially, our approach is data-agnostic: it requires only direct access to model weights, bypassing any need for training or validation data. The same vulnerability even extends to Mixture-of-Experts (MoE) language models, in which each token is routed through only a small subset of experts: for Qwen3-30B-A3B-Thinking, two sign flips into two different experts are sufficient to reduce accuracy from 78% to 0%.

---

*Equal contribution

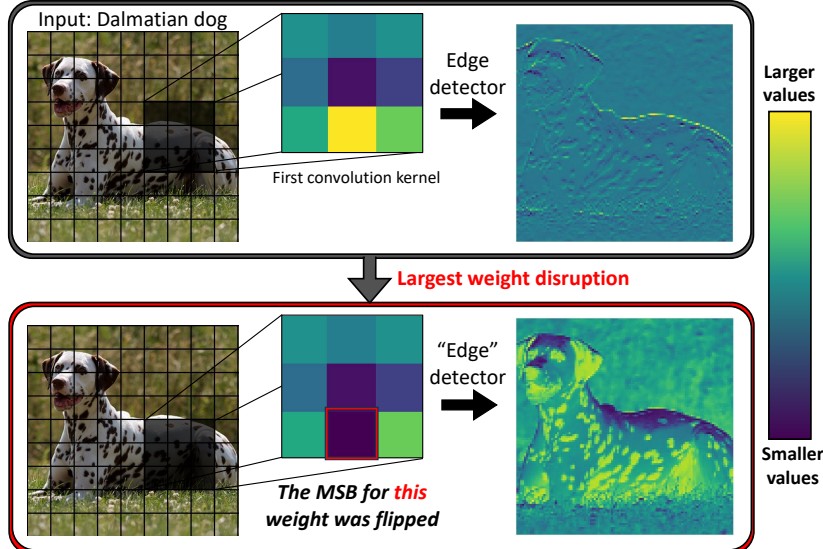

Figure 1: DNL applied to RegNetY-400MF's Radosavovic et al. (2020a) first convolution layer. The original (Sobel-like) kernel, used for horizontal edge detection, is shown above the flipped version obtained by changing just one high-magnitude weight's sign bit. Even this minimal alteration leads to a drastically different output feature map. This corrupted feature propagates through the model, undermining downstream representations and severely impairing the network's overall ability to recognize the Dalmatian.

Our method is deliberately lightweight. The forward-pass-free version, DNL requires no additional computational passes and ranks candidate weights using a magnitude-based heuristic guided by inductive bias and the flow of information through the network. We also propose an enhanced 1-pass attack, 1P-DNL, that uses a single forward and backward pass on random inputs to refine the selection of critical parameters. Both variants remain computationally inexpensive compared with prior weight-space attacks, yet still induce catastrophic failures across very different architectures and tasks.

Malicious actors can exploit the identified parameter vulnerability through multiple system layers, including file-system intrusions, firmware compromises, direct memory access (DMA) from compromised peripherals, or memory-level exploits. In each case, once attackers gain access to the model's parameters, they can flip a small number of carefully selected bits and trigger severe model failures. This lightweight approach requires no iterative optimization, thereby reducing the attacker's overhead while also increasing stealth.

For example, consider the implications for autonomous driving systems. Traditional adversarial attacks Madry et al. (2018); Goodfellow et al. (2015); Carlini & Wagner (2016) on such systems would involve manipulating the input pixels in real-time, requiring continuous communication with the vehicle and performing intensive gradient calculations to mislead the model. Physical adversarial attacks, such as placing adversarial stickers on street signs (e.g., as demonstrated in Wei et al. (2022)), demand direct access to the environment and are vulnerable to countermeasures like sensors on street signs or validation through additional traffic inputs. Other attacks targeting weights, such as Rakin et al. (2019); Park et al. (2021), rely either on the model's real data or on costly synthetic data generated from it, and demand an exhaustive search involving numerous forward and backward passes through the victim's model. Such attacks are impractical in real time because of their costly optimization process, and require the adversary to have full access to run the model as well as its data beforehand.

With our proposed method, even with limited access, an attacker could exploit any of the aforementioned hardware or software vulnerabilities and discreetly flip a small number of critical bits. By altering only a handful of parameters, often just one or two in the strongest cases, the attacker can critically degrade the model's perception, reasoning, and downstream decision-making, posing a far more potent threat to

the reliability of deployed systems. The minimal computational footprint and high impact of this attack make it exceptionally challenging to detect and mitigate in real-world deployments. Code is available at https://github.com/IdoGalil/maximal-brain-damage.

**Our main contributions are summarized as follows:**

**The DNL Attack:** We introduce the DNL attack, which exposes a severe vulnerability in DNNs by showing that heuristically flipping a small number of specific sign bits can catastrophically degrade model performance. This attack is entirely data-agnostic, requiring no knowledge of training data, domain-specific data, or synthetic inputs. Our method includes two lightweight variants: the "Pass-free" Attack, which operates without any additional computational passes, and the "1-Pass" Attack, which uses a single forward and backward pass with random inputs to refine the selection of critical parameters.

**Characterization of Critical Parameters:** We characterize the attributes that make certain parameters disproportionately vulnerable to bit flips, such as large magnitude and early-layer placement, while showing that the vast majority of parameters remain robust to random perturbations. We further distinguish which heuristics are generic (e.g., early-layer targeting) and which are architecture-specific (e.g., one flip per convolutional kernel).

**Extensive Evaluation Across Domains:** We validate our approach on image classification, object detection, instance segmentation, and reasoning large language models. In image classification alone, we evaluate 60 classifiers across diverse tasks and datasets, including 48 ImageNet models from the publicly available timm Wightman (2019) and Torchvision Marcel & Rodriguez (2010) repositories. Across all three domains, flipping only a very small number of targeted bits is sufficient to induce severe degradation.

**Defense Evasion:** We demonstrate that DNL circumvents various defenses, including binarization, redundancy-coding, and weight-scaling, underscoring the need for new protective strategies.

**Defense Mechanisms:** We leverage the insight gained from identifying critical parameters to propose efficient defenses. By selectively protecting only these most vulnerable parameters, models can become substantially more resilient to sign-flip attacks.

## 2 Problem Setup

Modern neural-network parameters may be stored in several numeric formats. For the floating-point formats relevant to this work (e.g., FP32, FP16, and bfloat16), each parameter has a dedicated sign bit, typically the most significant bit (MSB). Flipping this bit changes a weight from $\theta_i$ to $-\theta_i$, whereas flipping exponent bits changes its scale. For example, in IEEE 754 FP32, a value is represented as $(-1)^s \times 2^{(e-127)} \times \left(1 + \frac{m}{2^{23}}\right)$, and analogous decompositions hold for other floating-point formats as well.

We focus on a standard supervised learning scenario where a model $f_\theta$ is trained on a dataset with distribution $\mathcal{D}$. Let $\mathcal{X}, \mathcal{Y}$ be the input and label spaces, $(X, Y) \sim \mathcal{D}$, where $X \in \mathcal{X}$ and $Y \in \mathcal{Y}$. A trained model $f_\theta$ seeks to minimize the expected risk $\min_\theta \ \mathbb{E}_{(X,Y) \sim \mathcal{D}}\left[\mathcal{L}\big(f_\theta(X), Y\big)\right]$, where $\mathcal{L}$ is a loss function.

**Threat model.** The attacker has write access to the stored parameters, but no access to any kind of data, nor passing anything through the model. Concretely: (i) the attacker has no samples from $\mathcal{D}$ and thus no access to $P(X)$ or $P(Y)$; and (ii) the attacker cannot evaluate the model on any input (no forward or backward passes). Equivalently, for any input random variable $Z$ on $\mathcal{X}$ (including $Z = X$ or any synthetic/random data), the attacker has no access to $f_\theta(Z)$ or to $P\big(f_\theta(Z)\big)$ (for 1-Pass DNL we consider a slightly relaxed version where $f_\theta(z)$ could be observed for a single random input z, and allow a single forward pass and a single backward pass).

Despite this, the attacker can modify $\theta$ by flipping a small number of bits in its stored representation. Let bits$(\theta) \in \{0, 1\}^B$ be the $B$ memory bits encoding all entries of $\theta$ (e.g., IEEE-754). A *k-bit flip* chooses $k$

distinct bit indices $j_1, \ldots, j_k \in \{1, \ldots, B\}$ and produces parameters $\theta'_{(k)}$ with

$$\text{bits}\big(\theta'_{(k)}\big)_j = \begin{cases} 1 - \text{bits}(\theta)_j, & \text{if } j \in \{j_1, \ldots, j_k\}, \\ \text{bits}(\theta)_j, & \text{otherwise.} \end{cases}$$

We refer to $\theta'_{(k)}$ as the result of a $k$-bit-flip attack on $\theta$. To our knowledge, no prior method satisfies this restrictive threat model.

**Mechanisms enabling parameter bit flips.** The attacker can gain access to $\theta$ directly through software, firmware, or hardware-level exploits, namely Bit flip attacks Oriyano (2014).

Below, we outline several exploits that adversaries can leverage to execute malicious bit-flipping operations on model parameters.

A *rootkit* (Hoglund & Butler, 2006; Sparks & Butler, 2005; Rutkowska, 2007) is malicious software running with high-level (kernel or ring-0) privileges, allowing it to intercept or modify operations. Once installed, a rootkit can scan the system's memory or storage for the model's parameter files, then surgically flip bits in place. By concealing its processes and hooking system APIs, the rootkit can evade detection from common antivirus tools and monitoring systems, enabling stealthy, ongoing tampering with model parameters without triggering suspicious activity logs.

*Firmware exploits* (Hudson & Rudolph, 2015) (e.g., SSD/HDD controllers, GPU firmware, BIOS, or microcode patches) can give attackers privileged memory access or the ability to inject custom commands that flip bits in system memory or on storage media. By compromising firmware updates or exploiting known bugs, attackers can precisely manipulate parameter bits.

*DMA from untrustworthy peripherals* (Markettos et al., 2019) can read and write system memory without involving the CPU or the operating system's normal access controls. If attackers gain low-level access to a DMA device (e.g., via Thunderbolt or FireWire interfaces), they can directly overwrite targeted bits in protected memory regions.

*Rowhammer* (Kim et al., 2014b;a; Seaborn & Dullien, 2015) exploits the electrical interference between neighboring rows in modern DRAM modules. By rapidly accessing ("hammering") one row, an attacker causes bits in adjacent rows to flip, even without direct write permissions. Rowhammer attacks typically rely on high-frequency memory accesses that defeat standard refresh mechanisms; once carefully controlled, these flips can be directed at specific bit positions.

*GPU cache tampering* (Lipp et al., 2020; Tatar et al., 2018), which exploits a compromised kernel driver or malicious GPU code, can manipulate cache management routines to induce bit flips in stored parameters. Similar to Rowhammer's repeated DRAM accesses, continuously evicting and reloading specific cache lines may corrupt targeted parameters. Because GPU caches are often less scrutinized than CPU caches, this tampering can remain undetected, leading to stealthy yet severe degradation of model performance.

*Voltage/frequency glitching* (Murdock et al., 2020; Tang et al., 2017; Van der Veen et al., 2020; Frigo et al., 2018) manipulates the operating voltage or clock frequencies to induce computational errors. Certain voltage ranges can systematically cause specific bits to flip in registers or memory segments.

**Adversarial objective.** In all cases, the attacker's objective is to significantly degrade performance with minimal bit flips for stealth and practicality. We define the objective as: $\min_k \max \mathbb{E}_{(X,Y) \sim \mathcal{D}} \Big[ \mathcal{L}\big(f_{\theta'_{(k)}}(X), Y\big) \Big]$, where both finding minimal $k$ and flipping $k$ bits to produce $\theta'_{(k)}$ are discrete optimization problems.

In other words, the attacker's goal is to induce a significant performance drop while flipping only a handful of bits, both for stealth and practical reasons, as fewer corruptions are less likely to be detected and can be exploited by the mentioned hardware attacks. For instance, Rowhammer-based exploits (Kim et al., 2014b) typically induce only sporadic bit upsets in adjacent cells, making massive coordinated flips infeasible. By stealth, we mean that the modifications to the model weights (or inputs) are minimal—while the victim may observe a performance drop and even suspect an attack, the lack of an identifiable source makes it difficult to attribute the degradation or take effective countermeasures.

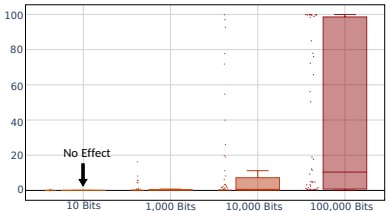 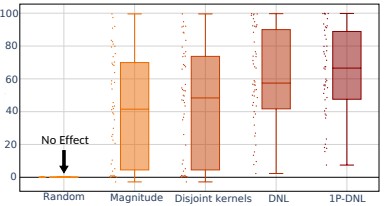 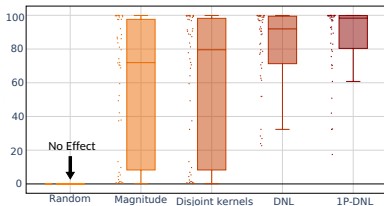

(a) $AR$ of randomly flipping sign bits on model performance.

(b) Comparison of $mAR_{10}$ across different strategies.

(c) Comparison of $AR(10)$ under different strategies.

Figure 2: Evaluation of model degradation under different sign-flip strategies across 48 ImageNet models.

As mentioned above, the attacker does *not* have access to any training or validation data, nor do they conduct extensive inference passes or iterative gradient-based searches. Such lightweight attacks are realistic in settings where the attacker's computational resources on the victim device are minimal, or where repeated forward/backward passes might raise suspicion. We therefore distinguish two scenarios: a *pass-free* attack, which uses no extra computation beyond reading and writing into the model weights (fitting our restrictive threat model), and a *1-pass* attack, which uses only a single forward (and backward) pass on a single random input, aiming for improved efficacy at a slight relaxation of the threat model. Both settings stand in contrast to existing approaches that require data samples and multiple optimization steps (see Section 5 for more details).

Although the attacker's objective can be framed as a discrete optimization problem—finding the smallest set of bits whose flips induce the greatest performance drop—exhaustive searches over millions of parameters are computationally infeasible in real-time and cannot scale with larger models. Instead, lightweight heuristics can pinpoint "critical" parameters while incurring minimal overhead. By leveraging inductive insights into how information flows through the network, an attacker can disrupt the most influential parameters without iterative optimization or a large dataset. This stands in contrast to data-driven or gradient-based methods, which demand multiple inference passes and raise both computational requirements and the risk of detection.

## 2.1 Accuracy Reduction Metrics

To measure the effect of bit flips, let $\theta'_{(k)}$ be the set of parameters obtained by flipping exactly $k$ bits in $\theta$. If $\mathrm{Acc}(\theta)$ is the model's original accuracy, we define: $\mathrm{AR}(k) = \frac{\mathrm{Acc}(\theta) - \mathrm{Acc}(\theta'_{(k)})}{\mathrm{Acc}(\theta)}$, which captures the drop in accuracy induced by $k$ flips. For a broader view, we also define:

$$\mathrm{mAR}(N) = \frac{1}{N} \sum_{k=1}^{N} \mathrm{AR}(k), \tag{1}$$

so that a single number can represent the model's overall vulnerability across different flip counts. Because practical hardware attacks often manage only a handful of flips, we mainly focus on small $k$ (e.g., $k \leq 10$).

## 3 Locating Models' Most Critical Parameters

**Targeting sign bits:** Considering the FP32 representation, exponent flips can alter a weight's magnitude, while flipping the most significant *sign* bit instantly switches a parameter from positive to negative (or vice versa). In the vision models that motivate our main method section, sign-bit flips provide the cleanest and usually strongest failure mode (see Appendix C), producing drastic changes in learned features as shown in Figure 1. For language models we additionally investigate exponent-bit attacks and find that they can be even more destructive, so the most harmful bit type is domain-dependent rather than universal. Localizing the sign bit in memory is straightforward (e.g., always the MSB), making it a simple target for adversaries. Various hardware-based studies show that repeated access patterns more reliably flip the *same* bit position across different addresses than arbitrarily chosen bits Kim et al. (2014b); Seaborn & Dullien (2015). Hence,

focusing on sign bits aligns with how hardware attacks often achieve consistent flips in a specific bit offset across multiple weights, increasing the chance of our targeted attack success rate.

Flipping random sign bits in a network's parameters typically has a negligible impact on performance. Indeed, our experiments (visualized in Figure 2a) show that for many architectures, flipping even up to $100,000$ bits does not reduce the accuracy consistently— indicating that most parameters are not "critical." These findings motivate a more targeted strategy to identify and flip only the most sensitive parameters.

**Magnitude-Based Strategy:** Drawing inspiration from the pruning literature, we first examine magnitude-based strategies. Just as magnitude pruning removes low-magnitude weights to minimize the impact on final predictions Frankle & Carbin (2018), we hypothesize that flipping the sign of *high-magnitude* parameters causes significant disruption. Formally, the parameter score function is defined as follows

$$\mathcal{S}(\theta_i) = |\theta_i| \tag{2}$$

As far as we are aware, this work is the first to evaluate the efficacy of a magnitude-based attack, a surprisingly simple yet powerful strategy that disrupts neural networks without data, optimization or prior knowledge. In Figure 2b, the second boxplot from the left, shows that focusing on the top-$k$ largest weights (in absolute value) significantly disrupts most evaluated models.

**One-Flip-Per-Kernel Constraint for Convolutional Models:** Empirical analyses of CNN filters (Krizhevsky et al., 2012; Zeiler & Fergus, 2014) highlight the importance of early-stage kernels (e.g., Gabor-like or Sobel-like) in extracting fundamental visual features. These studies reveal that flipping a single sign bit in a kernel can completely disrupt its feature extraction capability, altering the information the model relies on (see Figure 1 for the effect of sign flips on a real kernel). However, flipping multiple bits within the same kernel often merely changes its orientation or slightly modifies its functionality, rather than fully destroying the feature, as demonstrated in Figure 3. We observe this phenomenon consistently across multiple architectures. See Appendix B for examples.

One way to make the cancellation intuition more explicit is to look directly at how the kernel response changes. For a convolution kernel response $y = w^\top x$ on an input patch $x$, two sign flips at indices $i, j$ induce $\Delta y = -2(w_i x_i + w_j x_j)$. Thus, on a given patch, the second flip can partially offset the first whenever the two contributions have opposite signs. This is plausible in practice because natural-image patches are locally correlated and many early kernels have opposite-signed edge-detector lobes (Yosinski et al., 2014). Averaged over patches, the same effect appears in the mean-squared perturbation, which for patch covariance matrix $\Sigma$ gives

$$\mathbb{E}\big[(\Delta y)^2\big] = 4\big(w_i^2 \Sigma_{ii} + w_j^2 \Sigma_{jj} + 2w_i w_j \Sigma_{ij}\big).$$

When nearby patch entries are positively correlated ($\Sigma_{ij} > 0$) and large coefficients within the same kernel lie on opposite-signed lobes ($w_i w_j < 0$), the cross-term is negative, so the second flip can partially cancel the first instead of compounding it. This explains why spreading flips across kernels is more reliable than stacking them within one kernel.

To maximize damage in convolutional networks, we constrain the attack to flip exactly *one* bit per kernel, ensuring the disruption does not offset itself and affects a broader range of features. Given our focus on a small number of flips, distributing them across more kernels also helps amplify the overall impact. This heuristic is specific to convolutional filters and is not used in our transformer-based language-model experiments.

**Layer Selection** Beyond which parameters to flip, we also investigate *where* in the network to apply the attack. One might intuitively expect that targeting *final* layers—being closer to the classifier—would cause greater damage. However, our experiments reveal that in many architectures, early-layer manipulations are disproportionately damaging. Drawing on an analogy from neuroscience, early lesions (e.g., in the retina or optic nerve) can cause severe or total blindness (Kandel et al., 2000; Stewart et al., 2020; Essen et al., 1992). Similarly, flipping a single parameter in a fundamental feature detector (e.g., Sobel and Gabor filters) sends erroneous signals throughout subsequent layers, often leading to compounding error. Figure 1 illustrates this: a sign flip in a low-level "edge-detection" filter causes the network to misinterpret critical structural cues, compounding errors to later layers and severely degrading performance —more so than flips occurring in higher-level layers. Moreover, early convolutional filters encode generic edge and texture features; disrupting

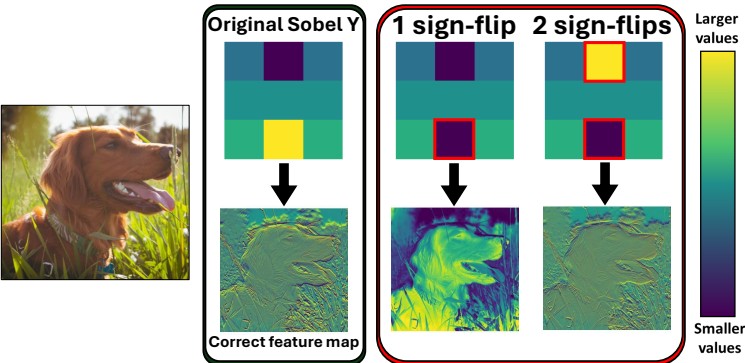

Figure 3: Horizontal edge detection filter (based on the Sobel Y filter) with one or two sign flips and their corresponding extracted features. With a single sign flip, the filter is severely disrupted, rendering it unable to detect edges effectively. However, with two bit flips, the resulting errors may partially offset each other, allowing the filter to retain some edge-detection capability and produce features similar to the original.

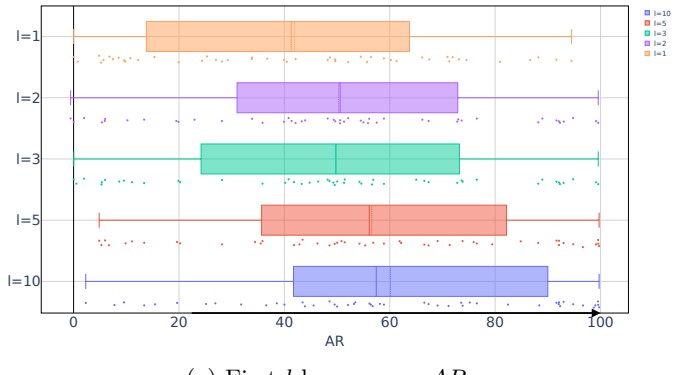

(a) First $l$ layers vs. $mAR_{10}$.

| Targeted | 1 | 2 | 3 | 5 |
|---|---|---|---|---|
| All Layers | .13 | .15 | .24 | .39 |
| First 100 | .19 | .24 | .42 | .48 |
| First 10 | **93.9** | **99.6** | 99.6 | **99.8** |
| First 5 | 93.9 | 99.6 | **99.7** | 99.7 |
| First 2 | 93.9 | 99.6 | **99.7** | 99.7 |
| First 1 | 59.5 | 53.7 | 82.4 | 88.5 |
| Last 10 | .01 | .04 | .06 | .17 |
| Last 5 | .03 | .19 | .46 | 1.14 |

(b) ShuffleNetV2 layer targets ($AR\%$).

Figure 4: Layer-specific vulnerability: global trend (left) and detailed case (right).

them degrades all downstream representations. This aligns with pruning evidence that early layers are disproportionately salient Frankle & Carbin (2018); Liu et al. (2019b). The same propagation intuition extends beyond CNNs: in transformer models, corruptions introduced in early blocks can similarly influence all later computations, motivating the early-layer targeting we also study for language models.

Interestingly, for most models evaluated, the largest parameters (*in absolute value*) tend to concentrate in these early layers. However, many models such as ShuffleNetV2 (Ma et al., 2018) exhibit a different pattern: their largest parameters are concentrated in later layers. As a result, naive attacks that always target the largest parameters—often located in the late layers of ShuffleNetV2—are less effective. Redirecting the attack to early layers, however, significantly amplifies the damage (see Table 4b for quantitative details).

**Theoretical motivation.** We can interpret our sign-flip attack through the same quadratic loss model long used to justify pruning criteria such as Optimal Brain Damage (OBD) LeCun et al. (1989b) and Optimal Brain Surgeon (OBS) Hassibi et al. (1992). For a trained network with parameters $\theta$ and loss $\mathcal{R}(\theta)$, a second-order Taylor expansion gives

$$\Delta\mathcal{R} \approx g^\top \Delta\theta + \tfrac{1}{2}\Delta\theta^\top H \Delta\theta,$$

where $g = \nabla_\theta \mathcal{R}(\theta)$ and $H$ is the Hessian. At convergence, $g \approx 0$, so curvature dominates Dong et al. (2020); Wang et al. (2019b). Flipping the sign of weight $\theta_i$ yields $\Delta\theta_i = -2\theta_i$, with all other coordinates unchanged,

giving under a diagonal-Hessian approximation:

$$\Delta \mathcal{R}_i \approx \tfrac{1}{2}(-2\theta_i)^2 H_{ii} = 2\,\theta_i^2 H_{ii}.$$

Thus, with a budget of $k$ flips, the greedy maximizer is the set of $k$ indices with largest $\theta_i^2 H_{ii}$. If $H_{ii}$ is approximately constant within a layer (empirically common in early convolutional layers and often a useful local approximation more generally), this reduces to choosing the $k$ largest $|\theta_i|$, precisely our *zero-pass* criterion. Equivalently, if $H \succeq \mu I$, then

$$\Delta \mathcal{R} \geq 2\mu \sum_{i \in S} \theta_i^2,$$

so picking the largest magnitudes maximizes a certified lower bound on loss damage.

Our *one-pass* variant (1P-DNL) refines this by replacing $H_{ii}$ with Gauss–Newton style estimates derived from gradients, recovering the classical Taylor saliency $\propto |\theta_i g_i|$ used in pruning (Molchanov et al., 2017; Lee et al., 2019; Wang et al., 2020; Tanaka et al., 2020). Hence, magnitude- and gradient-based flips correspond to adversarial analogues of the very same criteria known to predict weight importance.

**Early-layer targeting.** Our main support for early-layer targeting is empirical: restricting candidate flips to the first layers consistently increases attack impact (Figure 4a and Table 4b). A possible intuition is that a perturbation inserted earlier is processed by all subsequent layers. Under the standard Lipschitz composition bound, its worst-case amplification is at most $\prod_{\ell>1} L_\ell$, where $L_\ell$ is the Lipschitz constant of layer $\ell$ (Burago et al., 2001; Gouk et al., 2021; Virmaux & Scaman, 2018, Prop. 1.4.3).

**The Deep Neural Lesion Pass-free algorithm:** Based on these observations, we explore a simple heuristic algorithm that flips bits only in the first $l$ layers of a network (with $1 \leq l \leq 10$). Algorithm 1 summarizes our Pass-free attack for a given model, number of sign flips $k$, and layers $l$. We find that any $l$ in this range consistently degrades accuracy more than random or purely magnitude-based strategies (Figure 4a). We select $l = 10$ for simplicity and only consider the parameters of those layers as candidates for sign flips.

---

**Algorithm 1** Deep Neural Lesion (DNL) – Pass-free Attack

---

1: **Inputs:** Model parameters $\theta$, number of bits to flip $k$, number of layers $L$
2: $\theta_L \leftarrow$ parameters in the first $L$ layers of $\theta$
3: Sort $\theta_L$ in descending order by $|\theta_i|$
4: $\mathcal{K} \leftarrow$ top-$k$ entries of $\theta_L$
5: **For CNNs:** enforce at most one selected entry per kernel
6: **for** each $\theta_i$ in $\mathcal{K}$ **do**
7: $\quad \theta_i \leftarrow -\theta_i$ // flip sign bit
8: **end for**
9: **Output:** Modified parameters $\theta$

---

## 3.1 Enhanced Attack Using a Single Forward Pass

When a single forward (and backward) pass is within the attacker's budget, we propose an enhanced attack, called 1P-DNL, inspired by *gradient-based pruning* methods (LeCun et al., 1989a; Mozer & Smolensky, 1988; Lee et al., 2019; Wang et al., 2020; Tanaka et al., 2020). These methods typically assign a saliency or importance score to each parameter $\theta_i$ by measuring how altering that parameter (e.g., pruning or modifying it) would affect the network's loss or outputs. Although pruning and sign-flip attacks differ in goal, the underlying idea of identifying the "most critical" weights is similar.

**Hybrid Importance Score**

We define a hybrid importance scoring function that combines magnitude-based saliency with second-order information. This is not a new pruning criterion: the $|\theta_i|$ term matches magnitude-based pruning (Frankle & Carbin, 2018), while the second-order component follows Taylor/OBD-style saliency (LeCun et al., 1989a;

Hassibi et al., 1992); our contribution is to use this combination for adversarial parameter manipulation under a strict one-pass budget. Let $\alpha$ and $\beta$ be tunable coefficients controlling the relative weight of magnitude- and gradient-based terms. For a given parameter $\theta_i$,

$$\mathcal{S}(\theta_i) \;=\; \alpha\,|\theta_i| \;+\; \beta\left|\frac{\partial\mathcal{R}}{\partial\theta_i}\,\theta_i \;+\; \frac{1}{2}\,H_{ii}\,\theta_i^2 \;+\; \sum_{j\neq i} H_{ij}\,\theta_i\,\theta_j\right|, \tag{3}$$

where $H$ is the Hessian of $\mathcal{R}$ with respect to $\theta$. In our case, we let $\alpha = \beta = 1$, and define $\mathcal{R}(\theta)$ as the sum of model outputs on a random input (e.g., class scores for Gaussian image-like inputs in vision models, or logits induced by random token inputs in language models). Although the summation over $j \neq i$ captures inter-weight coupling, we approximate $H_{ij} = 0$ for $j \neq i$ (a common diagonal approximation in second-order pruning (LeCun et al., 1989a)), significantly reducing computation. Similarly, we replace $H_{ii}$ by $(\frac{\partial\mathcal{R}}{\partial\theta_i})^2$ (i.e., a Gauss-Newton like approximation), which further simplifies Hessian-based estimation.

- If $\frac{\partial\mathcal{R}}{\partial\theta_i} = 0$ and $H_{ii} = 0$, Eq. equation 3 reduces to: $\mathcal{S}(\theta_i) \;=\; \alpha\,|\theta_i|$, mirroring a simple magnitude-based saliency score (identical to Equation 2).

- If $\alpha = 0$, we recover a purely second-order (Optimal Brain Damage-like) approach:

$\mathcal{S}(\theta_i) \;=\; \beta\left|\frac{\partial\mathcal{R}}{\partial\theta_i}\,\theta_i \;+\; \frac{1}{2}H_{ii}\,\theta_i^2\right|$, which focuses on changes in $\mathcal{R}$ under small parameter perturbations.

Although one forward/backward pass on random data might be required to estimate $\mathcal{S}$, it remains significantly simpler than full data-driven optimization-based attacks (e.g., iterative gradient-based bit-flips). Figure 2b shows that incorporating second-order signals consistently amplifies the attack's damage compared to purely magnitude-based methods. Consequently, this hybrid scoring approach yields a more powerful single-pass sign-flip attack in scenarios where the attacker can run a forward and backward pass on the architecture, yet does not have access to the original training set. To summarize 1P-DNL, we refer the reader to Algorithm 2. Figure 2c shows the impact of all previously suggested methods with 10 sign flips. Both DNL and 1P-DNL cause most models to collapse, with 43 out of 48 models exhibiting an accuracy reduction above 60%. Finally, in Appendix D we compare 1P-DNL with various other 1-pass methods from the weight pruning literature to find critical parameters and find 1P-DNL the most potent.

## 4 Results Across Domains

We now evaluate DNL and 1P-DNL across three domains. Section 4.1 studies reasoning language models, where two sign flips into different experts already collapse Qwen3-30B-A3B and where exponent flips are even more destructive. Section 4.2 revisits image classification beyond ImageNet across additional datasets, and Section 4.3 shows that attacking only the backbone is enough to collapse object detection and instance-segmentation metrics. Unless stated otherwise, the main text focuses on sign-bit attacks, which provide the cleanest comparison across domains.

### 4.1 Language Models

We evaluate three reasoning LLMs—Qwen3-4B, Qwen3-30B-A3B, and Llama-3.1-Nemotron-Nano-8B (Qwen Team, 2025; NVIDIA, 2025)—on a fixed 50-question subset of MATH-500 derived from the MATH benchmark (Hendrycks et al., 2021). We score generations by answer accuracy using the benchmark's canonical verifier. The attack itself is the same DNL / 1P-DNL procedure, but without the convolution-specific one-flip-per-kernel constraint. In the sign-bit setting, the clean accuracies of the main targeted runs are 78% for Qwen3-30B-A3B, 86% for Qwen3-4B, and 94% for Nemotron Nano.

Several trends are worth highlighting. First, all three models are vulnerable to targeted sign flips, but the best layer scope is not universal: restricting the candidate set to the first five blocks is strongest for Qwen3-30B-A3B and Nemotron Nano, whereas Qwen3-4B is more vulnerable when all layers are considered. For compactness, Table 1 keeps only the first-five-block rows for Qwen3-30B-A3B and Nemotron Nano. In the available all-layer runs, Qwen3-30B-A3B still reaches 100.0% AR under DNL, but only after 7 flips instead of 2, while its all-layer 1P-DNL run never reaches 90% AR up to $k = 100$. Nemotron Nano shows the same

Table 1: Sign-bit attacks on reasoning LLMs, evaluated on the 50-question MATH-500 subset. Each attack column lists *# flips → AR (%)*. When a run reaches at least 90% relative accuracy reduction, we report the first such budget. Otherwise, we report a larger illustrative budget directly in the cell.

| Model | Targeted Layers | DNL Flips → AR (%) | 1P-DNL Flips → AR (%) |
|---|---|---|---|
| Qwen3-30B-A3B | First 5 blocks | 2 → 100.0 | 1 → 71.8,  4 → 100.0 |
| Qwen3-4B | First 5 blocks | 30 → 2.3 | 28 → 95.3 |
| | All layers | 14 → 100.0 | 4 → 95.3 |
| Nemotron Nano 8B | First 5 blocks | 32 → 100.0 | 17 → 100.0 |

<div>

**DNL, two sign flips**

```
I'm going to help you
with the solution.
I'm going to help you
with the solution.
I'm going to help you
with the solution.
...
```

**1P-DNL, four sign flips**

```
Hello,
I am a student,
I am a student,
I am a student,
...
```

</div>

Figure 5: Abridged generations from Qwen3-30B-A3B under sign-bit attacks on MATH-500. Left: DNL with two flips degenerates into repeated boilerplate. Right: 1P-DNL with four flips degenerates into repetitive text such as "I am a student". Both excerpts are abridged from raw outputs and illustrate why the same corruption mode is likely to transfer beyond MATH-500 to other generation benchmarks.

qualitative pattern: its all-layer DNL run reaches 93.2% AR only after 87 flips, and its all-layer 1P-DNL run never reaches 90% AR up to $k = 100$. Second, DNL alone is already sufficient to collapse Qwen3-30B-A3B with only two sign flips, while 1P-DNL already causes a 71.8% reduction with a single sign flip before reaching full collapse at four flips. The value of the single-pass refinement is even more visible on the more resistant models: for Qwen3-4B, 1P-DNL reaches severe degradation with 28 flips in the first-five-block setting and only 4 flips when all layers are available.

Targeted attacks are also far stronger than random sign flips. In the first-five-block setting, Qwen3-30B-A3B still retains 70% accuracy after 27 random flips, Qwen3-4B retains 80% accuracy after 100 random flips, and Nemotron Nano retains 92% accuracy after 100 random flips. Random sign flips therefore do not come close to the targeted collapse in Table 1. At the same time, their effect is less negligible than in our vision experiments, which is consistent with the hypothesis that autoregressive generation can compound even modest hidden-state corruption over time.

Qwen3-30B-A3B is especially striking because it is a Mixture-of-Experts model in which each token is routed through only a small subset of the available experts. The top two DNL sign flips target two different expert down-projection weights, one in layer 3 expert 82 and one in layer 1 expert 68, yet these two flips alone reduce accuracy from 0.78 to 0.00. This suggests that the disruption is not merely a local per-token routing failure. Rather, corrupting a small number of expert outputs appears able to poison latent token representations that then continue to propagate through the model. A complementary exponent-bit example supports the same interpretation: in a single-flip rank-check run, the attacked expert is used during prefill, and the response becomes gibberish immediately from the first generated tokens onward, even though the first several generated tokens do not route through that expert. This is consistent with corrupted hidden states propagating forward through attention, so that harming an expert that is not used on every generated token can still derail the entire response.

Representative corrupted generations for Qwen3-30B-A3B under sign-bit attacks are shown in Figure 5. These failures are not near-miss mathematical errors; the model quickly collapses into repetitive, nonsensical text. This is why we expect the same corruption mode to disrupt other generation-based benchmarks as well, not only MATH-500.

Table 2: Targeted sign-bit attacks on encoder-based text classification models fine-tuned on GLUE tasks (Wang et al., 2019a), including MRPC (Dolan & Brockett, 2005), QNLI (Rajpurkar et al., 2016), and SST-2 (Socher et al., 2013). We report the clean accuracy and the mean relative accuracy reduction over the ten flip budgets $k \in \{10, 20, 30, \ldots, 100\}$.

| Model | Task | Baseline | Mean AR over $k = 10, 20, \ldots, 100$ (%) |
|---|---|---|---|
| BERT | MRPC | 87.75% | 75.79 |
| | QNLI | 90.43% | 79.82 |
| | SST-2 | 93.16% | 82.43 |
| DistilBERT | MRPC | 84.80% | 75.15 |
| | QNLI | 86.13% | 78.7 |
| | SST-2 | 91.21% | 83.07 |
| RoBERTa | MRPC | 91.18% | 69.99 |
| | SST-2 | 94.34% | 77.44 |
| | QNLI | 92.19% | 75.42 |

We also evaluated exponent-MSB flips on the same LLMs. In the first-five-block setting, a single targeted exponent flip already reduces all three models to 0% accuracy under both DNL and 1P-DNL. Unrestricted targeting is nearly as destructive, although Qwen3-30B-A3B can require a few flips there. Random exponent flips are often highly destructive as well—for example, on Qwen3-30B-A3B a random exponent flip at $k = 1$ already reduces accuracy to 6%—which is consistent with exponent changes inducing extreme rescaling. Because this failure mode is both very strong and much less selective than sign attacks, we defer the detailed discussion to Appendix C.1.

**Text Encoders.** We also evaluated encoder-only language models fine-tuned for text classification on GLUE tasks (Wang et al., 2019a), using BERT (Devlin et al., 2019), DistilBERT (Sanh et al., 2019), and RoBERTa (Liu et al., 2019a) variants on MRPC, QNLI, and SST-2. As in the decoder-only language models in Section 4.1, we find that exponent-bit attacks can be more destructive than sign-bit attacks; however, for text encoders we focus here on sign-bit perturbations.

The results are summarized in Table 2. Across all nine encoder–task pairs, the attacks consistently cause severe degradation. Averaged over ten flip budgets $k \in \{10, 20, 30, \ldots, 100\}$, the relative accuracy reduction ranges from 69.99% to 83.07%, showing that the vulnerability is not limited to autoregressive generation. The strongest average degradation is observed on DistilBERT fine-tuned on SST-2, with a mean relative accuracy reduction of 83.07%, followed by BERT on SST-2 with 82.43%. Even the most robust setting we tested, RoBERTa on MRPC, still exhibits a mean relative accuracy reduction of 69.99%.

These results extend our cross-domain picture from decoder-only LLMs to encoder-based NLP models. The shared sensitivity of both model classes to bit-level perturbations suggests that even simple sign inversions can substantially degrade performance.

## 4.2 Image Classification

Beyond ImageNet, both DNL and 1P-DNL remain highly effective on DTD (Cimpoi et al., 2014), FGVC-Aircraft (Maji et al., 2013), Food101 (Bossard et al., 2014), and Stanford Cars (Krause et al., 2015). In Figure 6 and Figure 7, we plot the average accuracy reduction across EfficientNet-B0 (Tan & Le, 2019), MobileNetV3-Large (Howard et al., 2019), and ResNet-50 (He et al., 2015). In all four datasets, flipping one or two sign bits already leads to sharp collapse. Most notably, DNL yields $AR(5) \geq 85\%$ across all model/dataset combinations shown, while 1P-DNL reaches $AR(4) \geq 90\%$. Additional per-dataset plots are provided in Appendix E, as well as complete attack details per model in Appendix I.

**Impact of Model Size on Attack Success:** To assess whether model size influences the effectiveness of our attacks, we evaluate both DNL and 1P-DNL across five families of architectures with varying parameter counts: ResNet, RegNet, EfficientNet, ConvNeXt (Liu et al., 2022), and ViT (Dosovitskiy et al., 2020). The results, summarized in Figure 14 and Figure 15, reveal that model size does not exhibit a clear correlation

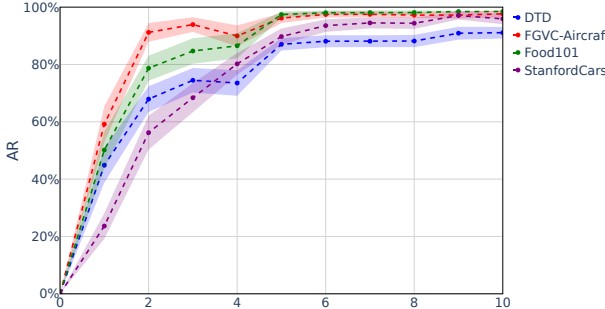 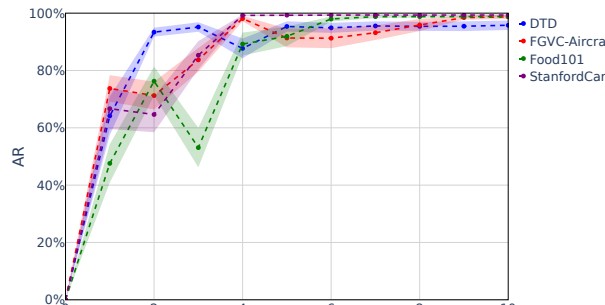

Figure 6: Averaged *AR* (%) of DNL over Efficient-NetB0, MobileNetV3-Large, and ResNet-50 vs number of sign flips. Each color represents a different dataset, confirming the impact of our pass-free attack on DTD, FGVC-Aircraft, Food101, and Stanford Cars.

Figure 7: Averaged *AR* (%) of 1P-DNL over EfficientNetB0, MobileNetV3-Large, and ResNet-50 vs number of sign flips. Each color represents a different dataset, confirming the fatality of our single-pass attack on DTD, FGVC-Aircraft, Food101, and Stanford Cars.

Table 3: Sign-bit attacks on COCO 2017 object detection and instance segmentation. DNL is applied only to the backbone. We report the baseline metric, the post-attack metric after $k = 1$ and $k = 2$ flips, and the corresponding relative reduction AR.

| Model / Backbone | Metric | Baseline | $k = 1$ | **AR(1)** | $k = 2$ | **AR(2)** |
|---|---|---|---|---|---|---|
| Mask R-CNN / ResNet-50 | bbox AP | 0.38 | 0.01 | 97.36 | 0.00 | 100.00 |
| | bbox AP50 | 0.59 | 0.03 | 94.93 | 0.00 | 100.00 |
| | segm AP | 0.35 | 0.00 | 100.00 | 0.00 | 100.00 |
| | segm AP50 | 0.56 | 0.01 | 98.21 | 0.00 | 100.00 |
| Mask R-CNN / ResNet-101 | bbox AP | 0.40 | 0.01 | 97.51 | 0.01 | 97.51 |
| | bbox AP50 | 0.61 | 0.03 | 95.12 | 0.02 | 96.75 |
| | segm AP | 0.36 | 0.00 | 100.00 | 0.00 | 100.00 |
| | segm AP50 | 0.58 | 0.01 | 98.28 | 0.00 | 100.00 |
| YOLOv8-seg | bbox AP | 0.33 | 0.05 | 83.66 | 0.05 | 86.33 |
| | bbox AP50 | 0.47 | 0.08 | 82.01 | 0.07 | 84.92 |
| | segm AP | 0.05 | 0.01 | 77.80 | 0.01 | 80.51 |
| | segm AP50 | 0.16 | 0.04 | 77.73 | 0.03 | 81.28 |

with attack susceptibility. Most models collapse at similar levels regardless of their scale, demonstrating that the attack is not confined to small networks.

## 4.3 Object Detection & Segmentation

We finally consider object detection and instance segmentation models evaluated on COCO 2017 (Lin et al., 2014). Here we attack only the backbone parameters and leave the task-specific heads untouched. We evaluate Mask R-CNN models with ResNet-50 and ResNet-101 backbones from torchvision (He et al., 2017; Lin et al., 2017; Marcel & Rodriguez, 2010), as well as YOLOv8-seg from Ultralytics (Jocher et al., 2023). We report average precision (AP): AP@[0.50:0.95] averages over IoU thresholds from 0.50 to 0.95, while AP@0.50 reports the same metric at IoU 0.50 only. We report these metrics for both bounding boxes (`bbox`) and instance masks (`segm`).

The results in Table 3 mirror the brittleness already seen in image classification. For both Mask R-CNN backbones, a single sign flip in the backbone already drives box AP to about 0.01 and mask AP to 0.00, with the two-flip setting effectively collapsing all reported metrics. YOLOv8-seg is somewhat more resilient, but even there one or two sign flips are sufficient to remove over 77% of both detection and segmentation

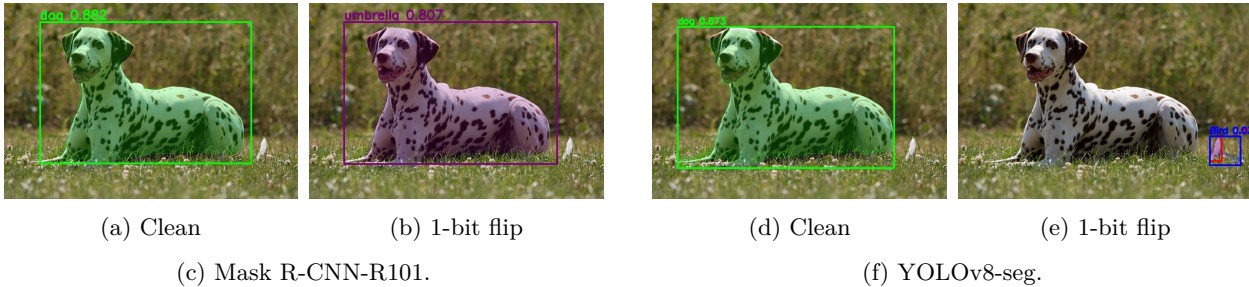

(a) Clean    (b) 1-bit flip          (d) Clean    (e) 1-bit flip

(c) Mask R-CNN-R101.              (f) YOLOv8-seg.

Figure 8: Qualitative comparison of dense prediction failures after a single targeted bit flip. The input image contains a dog. **Left two panels:** Mask R-CNN with a ResNet-101 backbone still segments the object with high fidelity after the attack, but assigns it the wrong semantic class. This is consistent with our attack protocol, which modifies only the backbone while leaving the task-specific heads untouched: localization and mask prediction can remain plausible even when the semantic representation has been corrupted. **Right two panels:** YOLOv8-seg detects the dog correctly in the clean setting, but after one bit flip it fails to recognize the dog altogether and instead hallucinates a bird detection on the tail. These examples illustrate two distinct failure modes induced by minimal parameter corruption: semantically incorrect yet well-localized prediction in Mask R-CNN, and complete object-level failure with hallucinated detection in YOLOv8-seg.

performance. These results show that the vulnerability is not limited to classification heads: corrupting a small number of backbone weights can derail downstream detection and segmentation as well. Figure 8 highlights two distinct qualitative failure modes under a one-bit attack: Mask R-CNN-R101 preserves coarse localization but assigns the dog an incorrect class, likely because our attack targets only the backbone, whereas YOLOv8-seg fails to detect the dog and hallucinates a bird on the tail.

# 5    Comparison to Other Weight Attacks

All prior methods rely on considerable optimization efforts, typically requiring multiple forward and backward passes through the network, and in most cases, access to data samples to compute gradients. Consequently, these attacks fall outside the scope of our restrictive threat model. Early works such as *Terminal Brain Damage* (TBD) Hong et al. (2019) illustrated how manipulating exponent bits could severely harm floating-point networks. However, TBD excludes sign bits, which in our vision-model experiments are often at least as devastating and frequently stronger at low flip budgets (see Appendix C). Other methods, including (Rakin et al., 2019; Yao et al., 2020), perform iterative gradient-based flips. For example, Rakin et al. (2019) requires multiple samples to compute gradients and can disrupt ResNet-50's accuracy by $\sim 99.7\%$ using 11 bit flips. Yao et al. (2020) similarly needs iterative optimization, reaching significant disruption at the cost of 23 flips.

Recent variants have attempted to relax data requirements. For instance, (Ghavami et al., 2021; Park et al., 2021) generate pseudo-samples or use partial data statistics to guide which bits to flip. Although they lessen the need for a large labeled dataset, they still rely on model feedback or approximate gradients. In contrast, our sign-bit flipping approach is lightweight, data-agnostic, and can degrade a large variety of vision networks by over 99.8% with few flips, while also transferring to reasoning LLMs and to object detection and instance segmentation models under the same broad threat model.

Table 4 compares bit-flip attacks on ImageNet-1K INT8 quantized models, and highlights how our approach differs from prior methods. BFA Rakin et al. (2019) and DeepHammer Yao et al. (2020) require iterative gradient-based searches and at least a few validation images. ZeBRA Park et al. (2021) drops the data requirement but still performs an optimization loop (and invests further compute into generating synthetic data). In stark contrast, our attacks can be carried out without data or optimization, and are straightforward to locate in memory, making them both feasible and devastating in real-world scenarios, yet equal or exceed prior art in accuracy reduction while using fewer bit flips. For example, 1P-DNL collapses ResNet-50 by 99.4% with a *single* sign flip.

Table 4: Bit-flip attacks on ImageNet-1K. Each cell lists *# flips → AR (%)*. OF = optimization-free, DA = data-agnostic. * best of 5 trials.

| Model | Method | OF | DA | # Flips → AR (%) |
|---|---|---|---|---|
| VGG-11 | BFA Rakin et al. (2019) | ✗ | ✗ | 17 → 99.7 |
| | ZeBRA Park et al. (2021) | ✗ | ✓ | 8 → 99.8 |
| | DNL | ✓ | ✓ | 3 → **99.9** |
| | 1P-DNL | ✓ | ✓ | 2 → 99.8 |
| ResNet-50 | DeepHammer Yao et al. (2020) | ✗ | ✗ | 23* → 75.4 |
| | BFA Rakin et al. (2019) | ✗ | ✗ | 1 → 5.5, 5 → 99.7 |
| | ZeBRA Park et al. (2021) | ✗ | ✓ | 1 → 7.7, 5 → 99.7 |
| | DNL | ✓ | ✓ | 1 → 6.6, 8 → 99.7 |
| | 1P-DNL | ✓ | ✓ | 1 → **99.4** |
| MobileNet-V2 | DeepHammer Yao et al. (2020) | ✗ | ✗ | 2* → **99.8** |
| | BFA Rakin et al. (2019) | ✗ | ✗ | 3 → 99.8 |
| | ZeBRA Park et al. (2021) | ✗ | ✓ | 2 → 99.7 |
| | DNL | ✓ | ✓ | 2 → 99.8 |
| | 1P-DNL | ✓ | ✓ | 2 → **99.9** |
| ViT-B/16@224 | BFA Rakin et al. (2019) | ✗ | ✗ | 5 → 30.1, 10 → 90.9 |
| | ZeBRA Park et al. (2021) | ✗ | ✓ | 5 → 5.1, 10 → 45.8 |
| | DNL | ✓ | ✓ | 5 → **99.3** |
| | 1P-DNL | ✓ | ✓ | 4 → 99.1 |

**LLM-specific weight attacks.** Recent work has extended adversarial weight manipulation from conventional DNNs to large language models, while also broadening the attack goals beyond generic accuracy collapse. Some studies emphasize stealth: *SilentStriker* (XU et al., 2025) aims to degrade task performance while preserving fluent, natural-looking generations. Other work focuses on scalability and model breadth: *FlipLLM* (Khalil & Hoque, 2025) formulates bit selection as a reinforcement-learning problem and demonstrates efficient attacks on both text-only and multimodal LLMs. More recently, *TFL* (Guo et al., 2026) studies targeted bit-flip attacks that steer selected prompts toward attacker-specified outputs while largely preserving utility on unrelated inputs. In parallel, Almalky et al. (Almalky et al., 2025) analyze how earlier bit-flip mechanisms transfer to LLMs, and Egashira et al. (Egashira et al., 2026) expose a related deployment-time threat in which malicious behavior is hidden in a model and activated after pruning. Relative to this emerging LLM literature, our focus remains intentionally more restrictive: the pass-free DNL setting is data-free and optimization-free, and even 1P-DNL requires only a single forward/backward pass on random inputs.

## 6 Defenses and Counter-Measures

**Selective Defense Against Sign-Flips.** A straightforward mitigation is to keep several full copies of every *sign* bit and take a majority vote at inference. Because an attacker would then have to corrupt most replicas *simultaneously*, the method is robust—but it multiplies memory and bandwidth.

A leaner alternative uses *error-correcting codes* (ECC) such as Hamming codes (Peterson & Weldon, 1972). Single-bit ECC can detect and fix isolated flips automatically, yet scaling to many parameters demands stronger (and costlier) codes. The key observation from Section 3 is that *only a tiny subset of sign bits is truly catastrophic*. Hence we can protect just those high-scoring weights—identified by DNL—either with bit replication or ECC, while leaving the vast majority of bits unguarded.

**Selective DNL Defense.** We tested DNL weight selection to defend against the iterative Bit-Flip Attack (BFA) (Rakin et al., 2019).

Table 5 reports the mean accuracy reduction after the attacker is allowed up to ten flips ($k \in [1, 10]$, three runs per $k$). Shielding only **0.001**% of parameters already halves BFA's impact on ResNet-18 and ResNet-50; guarding **1**% of parameters nullifies the attack on every model we tried.

While these experiments illustrate a simple defense, their greater significance lies in showing that DNL reliably identifies the most critical parameters—the very ones exhaustive BFA seeks to corrupt.

An expanded evaluation on different attack strategies with results on the sixteen most vulnerable architectures is provided in Appendix H.

Table 5: Effect of (DNL Defense) against BFA.

| Model | # Defended params | BFA AR(10) | Model | # Defended params | BFA AR(10) |
|---|---|---|---|---|---|
| ResNet-18 | No Defense | 88.87 | ResNet-50 | No Defense | 93.87 |
| | $\sim 0.001\%$ (100 params) | 58.83 | | $\sim 0.001\%$ (250 params) | 39.08 |
| | $\sim 1\%$ (100 K params) | **0.00** | | $\sim 1\%$ (250 K params) | **1.30** |
| MobileNet-V2 | No Defense | 99.90 | ViT-B/16@224 | No Defense | 82.30 |
| | $\sim 0.001\%$ (30 params) | 99.80 | | $\sim 0.001\%$ (900 params) | 40.51 |
| | $\sim 1\%$ (30 K params) | **44.30** | | $\sim 1\%$ (900 K params) | **0.21** |

**Evaluating DNL against existing defenses:** We evaluated representative defense strategies that have been proposed to prevent bit-flip attacks and found that DNL (and its 1-pass variant) either fully or largely bypass them.

**Encoding defenses.**

DeepNcode (Velcický et al., 2024) defends against parameter corruption by encoding each floating-point weight into a longer binary codeword with redundancy, typically using a codebook with Hamming distance $> 1$ between valid codewords. During inference, the stored codeword is decoded back into a floating-point value; if a bit flip occurs, the decoder maps the corrupted codeword to the nearest valid one, thereby correcting small errors.

However, this protection assumes the attacker cannot deliberately steer the corrupted codeword toward a *different valid codeword*. In a realistic *gray-box* setting—the attacker does not know the codebook, but can observe the resulting decoded values—we can exploit this decoding step. Specifically, by selectively flipping bits in the encoded representation, we search for the closest alternative codeword whose decoded value has the *opposite sign*. This effectively performs a sign flip *through* the encoding, bypassing the correction capability of the decoder.

**Weight-scaling defenses.** Weight-scaling (Fuengfusin & Tamukoh, 2024) multiplies all stored parameters by a constant $c$ and divides by $c$ at inference, damping additive perturbations by the factor $c$. Sign flips, however, are multiplicative ($\theta \mapsto -\theta$); after the rescaling they remain $-\theta$, exactly as in the undefended model: flip: $\theta \mapsto -\theta \implies$ defended: $\frac{-c\theta}{c} = -\theta$ Hence, the scaling defense has *no effect* on DNL, which is empirically confirmed by unchanged AR.

## 7 Concluding Remarks

This work exposes a fundamental vulnerability in deep neural networks: a cheap, heuristic, data-free attack that only requires access to stored weights can inflict severe damage across very different domains. We introduced a method for locating and flipping critical parameters, and showed that even without optimization or data it can catastrophically disrupt image classifiers, object detection and instance segmentation models, and reasoning language models. Building on the same critical-parameter analysis, we also proposed a targeted defense that selectively protects vulnerable sign bits and substantially improves robustness.

**Limitation.** DNL assumes that an adversary can directly modify a small number of stored parameters. In deployments where only part of the model is writable or addressable—for example because parameters are sharded, compartmentalized, or only partially exposed—the attack may be less effective, since a global search over all weights is no longer available.

Future work could explore partial-access threat models, as well as architectures, numeric formats, and training procedures that increase resistance to such lightweight attacks.

## 8 Acknowledgments

The research was partially supported by Israel Science Foundation, grant No 765/23

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

# A   Full cross-model comparison

Table 6 reports *all* flip-budget measurements we collected on full precision models, while Table 7 quantifies the corresponding computational savings. Both DNL and 1P-DNL remain the only attacks that are simultaneously **OF** (optimization-free) and **DA** (data agnostic), while achieving equal-or-higher accuracy reductions than optimization-based or data-requiring methods, often with fewer flipped bits. Note that Park et al. (2021) avoids requiring data by generating synthetic data from the victim model instead.

Table 6: Bit-flip attacks on ImageNet-1K. Each cell lists *# flips → AR (%)*. OF = optimization-free, DA = data-agnostic. * best of 5 trials.

| Model | Method | OF | DA | Flips → AR (%) |
|-------|--------|:--:|:--:|:--------------:|
| AlexNet | BFA Rakin et al. (2019) | ✗ | ✗ | 17 → 99.6 |
| | DNL | ✓ | ✓ | 10 → 88.5,  17 → 98.2 |
| | 1P-DNL | ✓ | ✓ | 10 → 93.2,  17 → 98.2 |
| VGG-11 | BFA Rakin et al. (2019) | ✗ | ✗ | 17 → 99.7 |
| | ZeBRA Park et al. (2021) | ✗ | ✓ | 8 → 99.8 |
| | DNL | ✓ | ✓ | 3 → **99.9** |
| | 1P-DNL | ✓ | ✓ | 2 → 99.8 |
| ResNet-50 | DeepHammer Yao et al. (2020) | ✗ | ✗ | 23* → 75.4 |
| | BFA Rakin et al. (2019) | ✗ | ✗ | 1 → 5.5,  5 → 99.7 |
| | ZeBRA Park et al. (2021) | ✗ | ✓ | 1 → 7.7,  5 → 99.7 |
| | DNL | ✓ | ✓ | 1 → 6.6,  5 → 40.4,  8 → 99.7 |
| | 1P-DNL | ✓ | ✓ | 1 → **99.4** |
| MobileNet-V2 | DeepHammer Yao et al. (2020) | ✗ | ✗ | 2* → **99.8** |
| | BFA Rakin et al. (2019) | ✗ | ✗ | 3 → 99.8 |
| | ZeBRA Park et al. (2021) | ✗ | ✓ | 2 → 99.7 |
| | DNL | ✓ | ✓ | 2 → 99.8 |
| | 1P-DNL | ✓ | ✓ | 2 → **99.9** |
| Inception-V3 | BFA Rakin et al. (2019) | ✗ | ✗ | 3 → 99.8 |
| | ZeBRA Park et al. (2021) | ✗ | ✓ | 3 → 99.8 |
| | DNL | ✓ | ✓ | 2 → **99.8** |
| | 1P-DNL | ✓ | ✓ | 3 → 99.1 |
| ViT-B/16@224 | BFA Rakin et al. (2019) | ✗ | ✗ | 5 → 30.1,  10 → 90.9 |
| | ZeBRA Park et al. (2021) | ✗ | ✓ | 5 → 5.1,  10 → 45.8 |
| | DNL | ✓ | ✓ | 5 → **99.3** |
| | 1P-DNL | ✓ | ✓ | 4 → 99.1 |

Table 7: Approximate computational costs for different bit-flip attacks. Here, $\theta$ is the number of parameters in the model, $k$ is the number of flipped bits, $m$ is the (mini-)batch size used for gradient or scoring, and $B$ is the number of candidate bits evaluated in each iteration. All complexities assume that a forward/backward pass scales on the order of $\mathcal{O}(\theta \times m)$.

| Method | Description | Complexity |
|--------|-------------|:----------:|
| **BFA** Rakin et al. (2019) | Iterative gradient-based search; each flip requires scoring multiple bits | $\mathcal{O}(k \times B \times \theta \times m)$ |
| **DeepHammer** Yao et al. (2020) | Chain-based iterative search for each flip | $\mathcal{O}(k \times B \times \theta \times m)$ |
| **ZeBRA** Park et al. (2021) | Zero real-data, but still repeated forward/backward passes per flip | $\mathcal{O}(k \times B \times \theta \times m)$ |
| **DNL (Ours)** | **Pass-free**; select bits by magnitude only | $\mathcal{O}(\theta) + \mathcal{O}(k)$ |
| **1P-DNL (Ours)** | **Single-pass**; one forward/backward pass | $\mathcal{O}(\theta) + \mathcal{O}(k)$ |

# B   One Flip Per Kernel Constraint Examples

*MobileNetV3-Large* (Howard et al., 2019): Applying our magnitude-based method for $k = 2$ selects the second highest-magnitude weight for sign flipping, which in this case belongs to the third convolutional layer, and results

in a significant accuracy drop to $AR(2) = 81.31$. Adding another magnitude-based weight results in a flip within the same kernel that reduces the degradation to $AR(3) = 46.97$, partially offsetting the attack. However, flipping the next highest-magnitude parameter from a different kernel instead raises the accuracy reduction dramatically to $AR(3) = 94.0$.

*RegNet-Y 16GF* (Radosavovic et al., 2020b): In the second convolutional layer, several of the top 10 highest-magnitude weights reside in the same kernel. Flipping the sixth highest weight yields $AR(6) = 74.2$, while also flipping the seventh highest weight (in the same kernel) improves accuracy to $AR(7) = 66.5$, rather than compounding the damage.

## C   Vision Models: Sign-Bit vs. Exponent-Bit Flips

While most prior work Section 5 focused on flipping exponent bits, our *vision* experiments show that sign-bit flips typically cause greater disruption per flip. Table 8 compares the impact of flipping the sign bit versus the most significant exponent bit on representative image models. Exponent-bit flips are still effective, as they can sharply alter weight magnitudes, but sign-bit flips are usually the more consistently destructive choice at low budgets in vision. This observation should be contrasted with the language-model results below, where exponent-bit attacks can be stronger.

Table 8: Sign-bit vs. exponent-bit flips ($k = 10$). AR: accuracy reduction; $mAR_{10}$: mean AR over 1–10 flips on full precision models.

| Model | Sign AR(10)/ mAR(10) | Exp. AR(10) / mAR(10) |
|---|---|---|
| VGG-11 | **91.8/91.6** | 53.89/ 34.48 |
| ResNet-18 | 70.63/38.6 | 99.9/ 99.8 |
| ResNet-34 | 56.3/76.3 | 99.9/92.7 |
| ResNet-50 | **99.7**/52.7 | 70.94/**65.48** |
| MobileNet-V2 | 99.85/92.17 | 99.86/92.0 |
| MobileNet-V3 | **99.42/88.37** | 91.87/70.46 |
| Inception-V3 | 96.8/52.1 | **99.9/99.9** |
| RegNetY-16GF | **82.87/61.9** | 78.75 / 48.6 |
| EfficientNet-B0 | **95.80 / 77.7** | 37.38 / 22.53 |
| ViT-B/16@224 | **99.84/99.52** | 82.38/47.27 |
| AlexNet | 88.5 / **43.45** | **99.8**/36.03 |

### C.1   Language Models: Exponent-Bit Flips

Language models exhibit a different pattern than vision models. In the first-five-block setting, a *single targeted exponent flip* already reduces all three reasoning LLMs to 0% accuracy under both DNL and 1P-DNL. Unrestricted targeting remains nearly as destructive, although Qwen3-30B-A3B can require a few flips there. Random exponent flips are also often severe: in the same first-five-block setting, a single random exponent flip already drops Qwen3-30B-A3B to 6% accuracy.

The strongest qualitative exponent example we inspected is again Qwen3-30B-A3B. A single rank-check exponent flip in `model.layers.3.mlp.experts.82.down_proj.weight` already collapses accuracy to 0%, introduces one non-finite parameter, and produces obviously corrupted multilingual/gibberish text. The attacked expert is used during prefill, yet the response becomes gibberish immediately from the first generated tokens onward, even though the first several generated tokens do not route through that expert. This supports the same error-compounding hypothesis raised in the main text: once the hidden state has been corrupted, the damage can propagate forward through attention even when the attacked expert is not used on every generated token. Over the full inspected example, the attacked expert is still routed on only 4.14% of tokens overall.

In the encoder setting, the effect is again severe, although less uniformly catastrophic than for the reasoning LLMs discussed above. For the AR(10) results, RoBERTa reaches 63.94 on SST-2, 49.99 on QNLI, and 65.32 on MRPC; BERT-base reaches 50.11, 49.03, and 63.97 on the same tasks; and DistilBERT reaches 49.04, 46.49, and 62.72, respectively. Across these examples, MRPC appears most vulnerable, while SST-2 and QNLI still exhibit substantial degradation.

One likely reason for the strength of exponent attacks is that they alter the exponent field itself and can therefore induce extreme rescaling rather than a simple sign inversion. By contrast, a sign-bit attack leaves the exponent and mantissa untouched and merely negates the stored value. This strong dependence on floating-point format suggests that quantized models may behave differently under exponent attacks, which we leave for future work.

## D  Weight Score Ablation

We evaluate several parameter scoring functions from the pruning literature and compare their effectiveness in identifying high-impact weights for sign-flip attacks. As shown in Figure 9, we measure the mean accuracy reduction $mAR_{10}$ across 48 ImageNet models under the following scoring functions:

- **Magnitude-based:** $S(\theta_i) = |\theta_i|$.
- **GraSP:** $S(\theta_i) = |\theta_i \odot Hg|$, following the gradient-flow preservation principle of Wang et al. (2020) where $Hg$ is the hessian vector product.
- **GraSP (Gauss-Newton Approx.):** Similar to GraSP but approximates the Hessian $H$ with the square of first-order gradients.
- **SynFlow:** $S(\theta_i) = |g \odot \theta_i|$, akin to gradient *times* weight.
- **Optimal Brain Damage (OBD):** $S(\theta_i) \approx \frac{1}{2}\theta_i^t H_{ii}\theta_i$ (LeCun et al., 1989a).
- **Hybrid (Ours):** As we define in Equation (3) with and without second order term.

where $g = \frac{\partial \mathcal{R}}{\partial \theta_i}, H = \frac{\partial^2 \mathcal{R}}{\partial \theta_i^2}$.

We observe that certain models are vulnerable to second-order-based scores (e.g., OBD) even when they prove more resilient to pure magnitude-based attacks. Nevertheless, other architectures appear more robust against OBD or GraSP while showing larger drops under magnitude-based score. Motivated by these mixed results, our hybrid score combines both magnitude and gradient terms. This blend consistently identifies critical weights even in cases where either component alone fails to degrade accuracy. Overall, the hybrid approach delivers the most reliable performance drop across the tested models.

## E  Additional Datasets Evaluation

Figures 10, 11, and 12 analyze individual dataset results on these three popular classifiers. Each shows a steep drop in accuracy with very few sign flips, highlighting the generality of the attack. Notably, although these models differ in architecture and capacity, they all exhibit severe degradation once our detected sign bits are flipped. This finding reinforces that our method targets fundamental weaknesses in DNN representations rather than exploiting quirks of a specific network or dataset.

## F  1P-DNL Algorithm

Similar to Algorithm 1, Algorithm 2 shows the algorithm of 1P-DNL.

**Seed sensitivity.**  DNL is deterministic: for a fixed model, layer budget, and flip budget $k$, it selects the same weights and produces the same accuracy reduction on every run. For 1P-DNL, the only stochasticity is the single random input used to compute the score. On a representative subset of architectures (ConvNeXt-B, RegNetY-400MF, ResNet-50, EfficientNet-B0, and ViT-B/16), repeating this step over 10 random seeds yields a standard deviation of 0.02 in accuracy reduction, which is negligible relative to the induced drops.

## G  Defenses

**More existing defenses**

**Binarization.**  Binary-weight networks Courbariaux et al. (2016); Liu et al. (2020); Xu et al. (2021); Rozen et al. (2022) are often assumed to be naturally resilient to weight perturbations He et al. (2020); Rakin et al. (2021), yet flipping a sign bit still inverts the weight. We show the results on a binarized ResNet-18 in Table 9, confirming that binarization alone offers negligible protection.[1]

---

[1]Results reproduced with the RA-BNN recipe of Rakin et al. (2021).

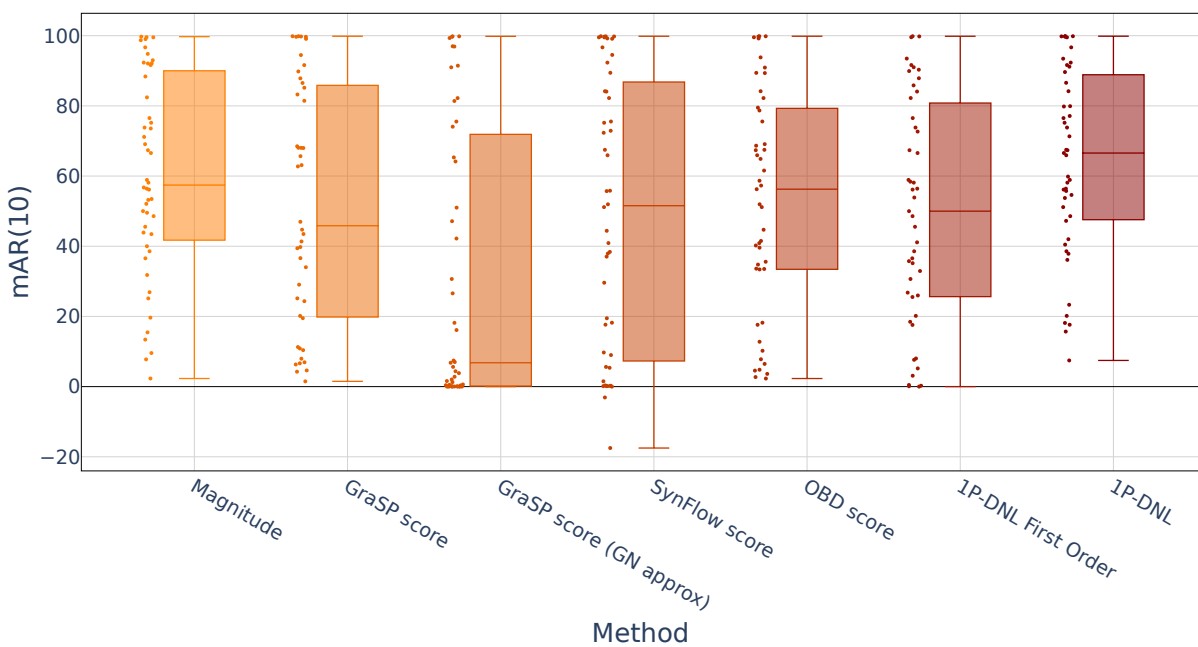

Figure 9: Comparison of $mAR_{10}$ across different weight score functions for the model parameters applied to 48 ImageNet models.

---

**Algorithm 2** 1P-DNL – Single-Pass Attack

---

1: **Inputs:** Model $f_\theta$, number of bits to flip $k$, number of layers $L$
2: $X \leftarrow$ random input (e.g., Gaussian noise or random tokens)
3: $\mathcal{R}(\theta) \leftarrow \sum_i f_\theta(X)[i]$    // e.g., sum of logits
4: $g \leftarrow \nabla_\theta \mathcal{R}(\theta)$    // one backward pass
5: $\theta_L \leftarrow$ parameters in the first $L$ layers of $\theta$
6: **for** each $\theta_i$ in $\theta_L$ **do**
7:     Approx. Hessian diagonal by Gauss–Newton: $H_{ii} \approx [g_i]^2$
8:     $\mathcal{S}(\theta_i) \leftarrow |\theta_i| + \left|\theta_i\, g_i + \frac{1}{2}\,\theta_i^2\, H_{ii}\right|$
9: **end for**
10: Sort $\theta_L$ in descending order by $\mathcal{S}(\theta_i)$
11: $\mathcal{K} \leftarrow$ top-$k$ entries of $\theta_L$
12: **For CNNs:** enforce at most one selected entry per convolutional kernel
13: **for** each $\theta_i$ in $\mathcal{K}$ **do**
14:     $\theta_i \leftarrow -\theta_i$    // flip sign bit
15: **end for**
16: **Output:** Modified parameters $\theta$

---

Table 9: $AR(\cdot)$ Targeting Binary ResNet-18 with DNL

| AR(1) | AR(2) | AR(3) | AR(5) | AR(10) |
|-------|-------|-------|-------|--------|
| 0.14  | 12.90 | 60.71 | 90.35 | 96.50  |

## H   Selective Defense Against Sign-Flips: Additional Setups

Following Section 6, to quantify this selective defense, we tested it on 16 particularly vulnerable networks, each suffering at least a 50% accuracy reduction ($AR(100,000) \geq 50\%$) when 100K random parameters were flipped. We

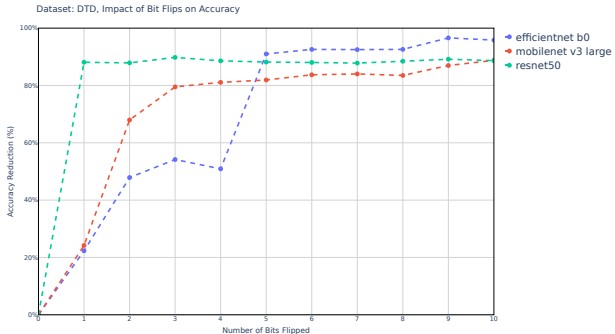

Figure 10: *AR* (%) on DTD dataset (Cimpoi et al., 2014) with varying number of sign flips over popular image encoders.

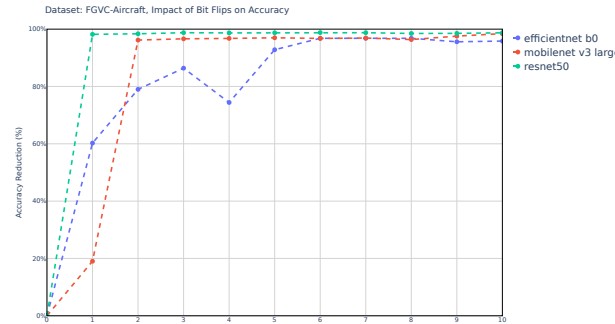

Figure 11: *AR* (%) on FGVC Aircraft dataset (Maji et al., 2013) with varying number of sign flips over popular image encoders.

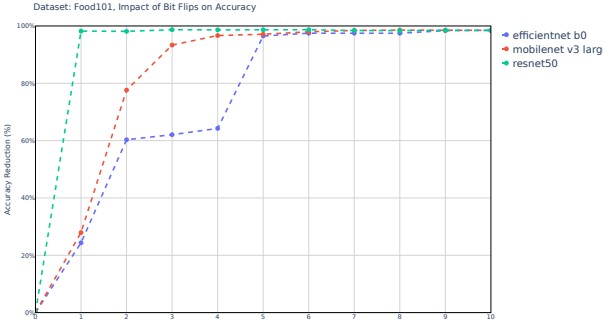

Figure 12: *AR* (%) on Food101 dataset (Bossard et al., 2014) with varying number of sign flips over popular image encoders.

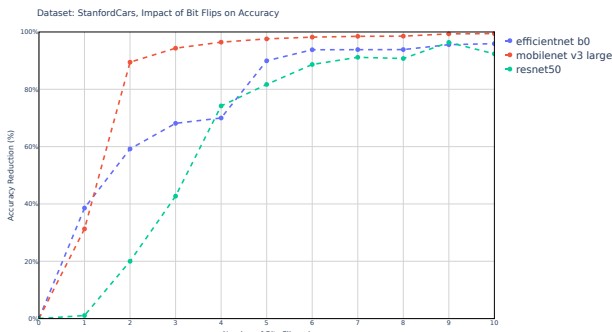

Figure 13: *AR* (%) on Stanford Cars dataset (Krause et al., 2015) with varying number of sign flips over popular image encoders.

use this large-scale random flip as a strong, non-specific stress test that is not tied to our own scoring method, ensuring that the defense remains robust to other score-based sign bit flips. We then varied the fraction of protected sign bits from 1% to 20%, focusing on the largest weights in absolute value. As expected, even a modest level of protection dramatically reduced the damage inflicted by sign-flip attacks. Moreover, as illustrated in Figure 16, selectively safeguarding this small subset of sign bits mitigates the impact from sign bit flip attacks, as reflected from the stress test proposed above, demonstrating that partial protection of critical parameters offers a practical and effective defense In Appendix G, we show that naive defense mechanisms would have been unsuccessful in defending against sign flips.

In addition to selectively protecting the most impactful sign bits, we also tested a baseline defense that shields a randomly chosen subset of bits at different coverage levels. Figure 17 shows that even when 20% of the sign bits are randomly protected, the network remains highly vulnerable under 100k random sign flips. This stands in stark contrast to protecting only a small fraction of critical sign bits (e.g., the largest-magnitude weights), which can substantially preserve accuracy. The results underscore that which bits get protected is more important than how many.

# I    Full ImageNet Model Tables

We report the full per-model accuracy reduction (AR) curves for all 48 ImageNet classifiers evaluated in this work in Tables 10 and 11. These tables complement the aggregate statistics presented in the main text (e.g., Figure 2) by providing detailed, model-level behavior across different flip budgets $k \in \{1, \ldots, 10\}$.

Consistent with the aggregate results, the majority of models exhibit severe degradation under targeted sign-bit flips. In particular, many architectures (e.g., MobileNet, MnasNet, ViT, and VGG families) reach near-complete collapse

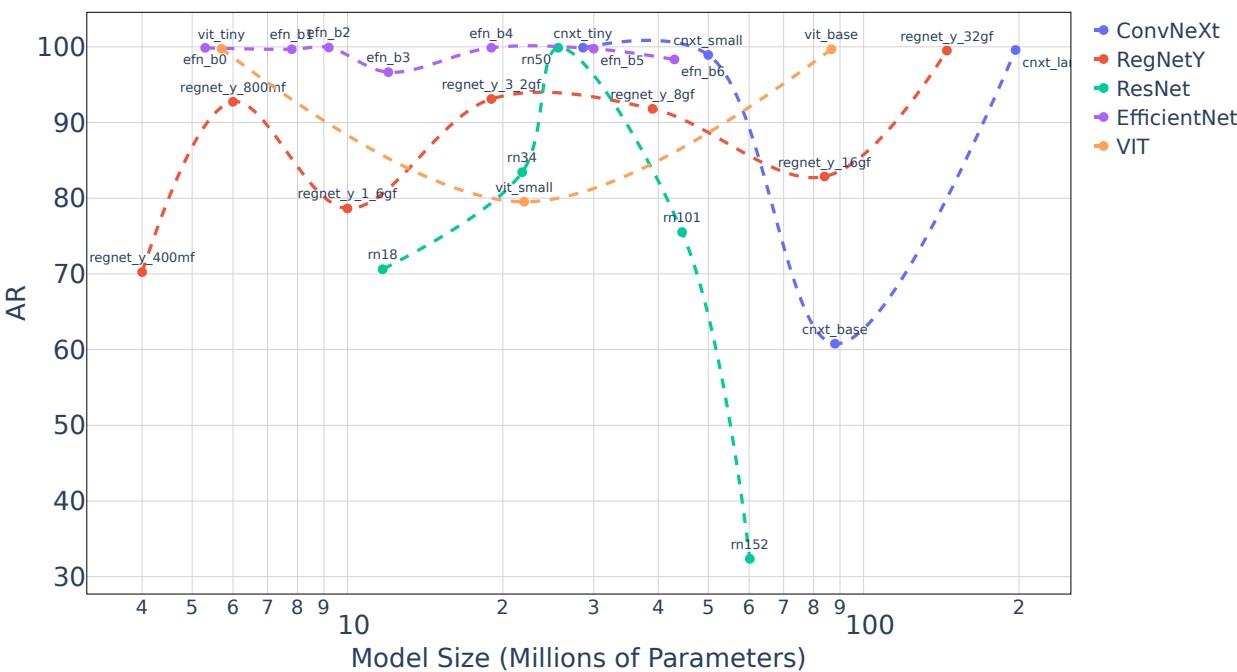

Figure 14: *AR* reported across five model families of varying capacities under 1P-DNL attack. The similar vulnerability levels suggest that model size alone does not mitigate sign-flip attacks.

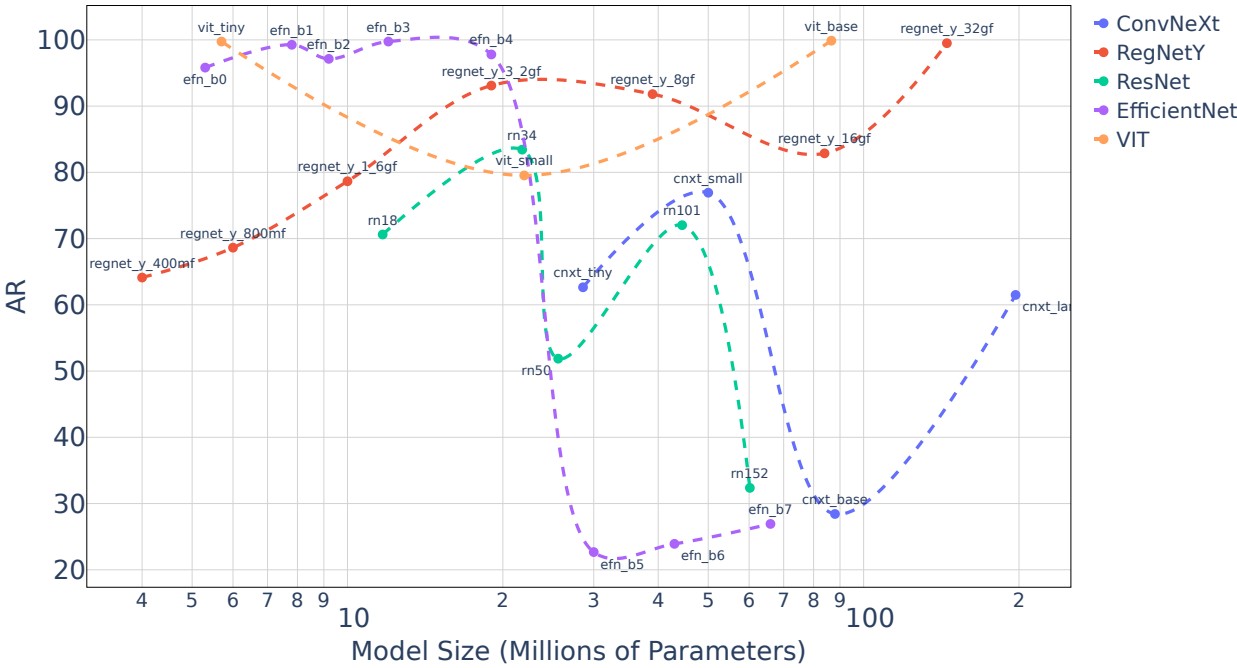

Figure 15: *AR* reported across five model families of varying capacities under DNL attack.

($AR \approx 100\%$) with fewer than 5–10 flips. This supports the claim that only a handful of carefully selected parameters are sufficient to disrupt modern neural networks.

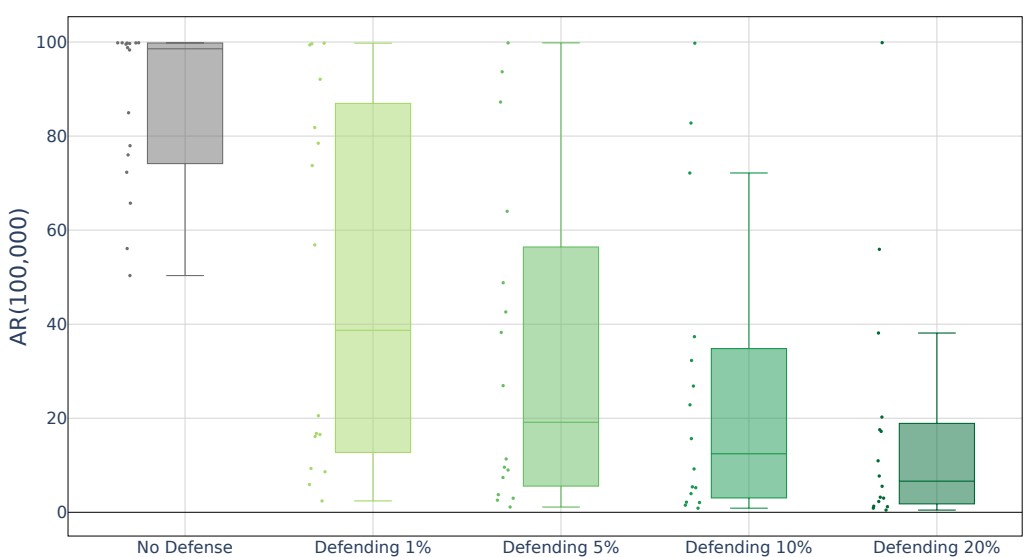

Figure 16: $AR(100,000)$ under 100k random sign flips, with selective protection on varying fractions of the most vulnerable parameters (ranked by DNL). Even partial coverage of high-scoring parameters substantially improves robustness.

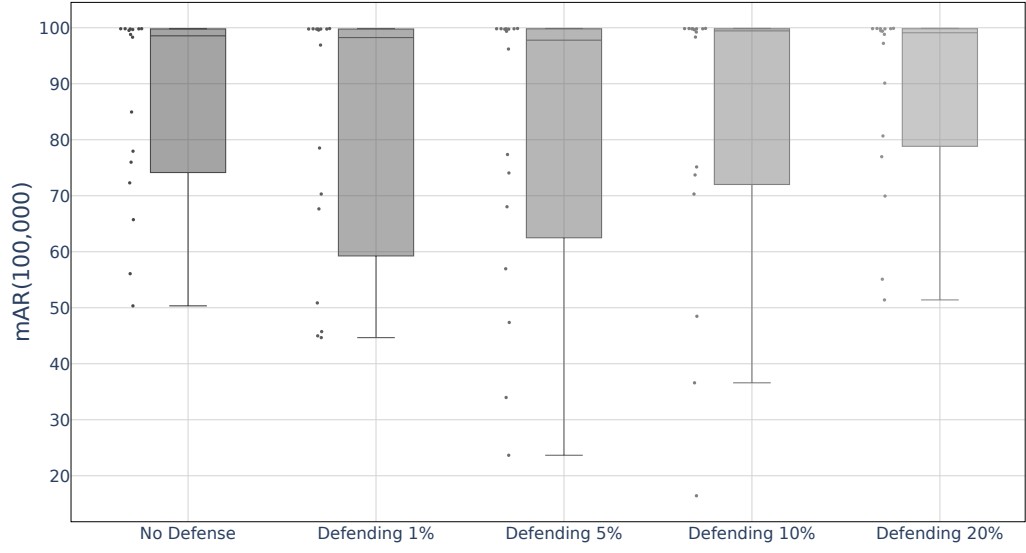

Figure 17: $AR(100,000)$ under 100k random sign flips, with random subsets (1%, 5%, 10%, and 20% coverage) of sign bits protected. Unlike Figure 16, where shielding the most vulnerable bits significantly reduces damage, uniform random selection offers little resilience, as even 20% coverage barely mitigates the attack.

| Model | Base Acc. | AR(1) | AR(2) | AR(3) | AR(4) | AR(5) | AR(6) | AR(7) | AR(8) | AR(9) | AR(10) | Avg. |
|---|---|---|---|---|---|---|---|---|---|---|---|---|
| alexnet | 56.51 | 6.2 | 12.8 | 13.3 | 17.0 | 45.1 | 50.8 | 49.9 | 67.5 | 83.4 | 88.5 | 43.5 |
| convnext base@fb in1k | 83.83 | 0.7 | 5.3 | 6.3 | 12.7 | 12.5 | 19.9 | 20.0 | 25.0 | 23.7 | 28.4 | 15.5 |
| convnext large@fb in1k | 84.29 | 0.7 | 0.6 | 1.1 | 1.9 | 4.4 | 4.5 | 56.0 | 59.5 | 61.1 | 61.5 | 25.1 |
| convnext small@fb in1k | 83.14 | 5.5 | 40.7 | 44.0 | 46.2 | 57.9 | 57.9 | 59.6 | 70.3 | 73.0 | 76.9 | 53.2 |
| convnext tiny@fb in1k | 82.06 | 5.4 | 23.0 | 31.6 | 32.9 | 38.7 | 60.7 | 62.6 | 62.2 | 60.7 | 61.1 | 43.9 |
| efficientnet b0 | 77.69 | 23.0 | 48.5 | 52.7 | 41.6 | 76.7 | 93.0 | 93.0 | 93.0 | 94.1 | 95.8 | 71.2 |
| efficientnet b1 | 77.43 | 5.0 | 11.1 | 35.4 | 48.5 | 98.1 | 98.1 | 98.1 | 98.0 | 99.2 | 99.3 | 69.1 |
| efficientnet b2 | 79.54 | 0.0 | 0.0 | 7.3 | 34.9 | 51.8 | 87.3 | 95.9 | 96.2 | 96.9 | 97.1 | 56.7 |
| efficientnet b3 | 81.49 | 32.9 | 49.5 | 47.7 | 99.7 | 99.6 | 99.6 | 99.3 | 98.6 | 98.6 | 98.6 | 82.4 |
| efficientnet b4 | 82.76 | 0.4 | 0.5 | 0.8 | 1.6 | 2.0 | 4.6 | 8.3 | 8.6 | 9.4 | 97.8 | 13.4 |
| efficientnet b5 | 85.89 | 0.0 | 0.9 | 3.7 | 4.3 | 4.2 | 4.4 | 7.1 | 8.0 | 22.7 | 22.2 | 7.8 |
| efficientnet b6 | 83.83 | 30.0 | 45.0 | 55.0 | 70.0 | 85.0 | 92.0 | 95.0 | 96.0 | 96.5 | 97.0 | 76.2 |
| efficientnet b7 | 84.4 | 0.0 | 0.5 | 1.5 | 3.0 | 5.0 | 7.0 | 10.0 | 15.0 | 20.0 | 25.0 | 8.7 |
| googlenet | 69.78 | 87.3 | 87.3 | 85.6 | 89.6 | 91.2 | 95.0 | 98.0 | 98.3 | 98.8 | 98.8 | 93.0 |
| inception v3 | 77.30 | 1.7 | 2.0 | 5.9 | 21.4 | 56.9 | 74.9 | 81.5 | 89.2 | 90.6 | 96.8 | 52.1 |
| mnasnet0 5 | 67.73 | 97.0 | 99.7 | 99.9 | 99.9 | 99.9 | 99.8 | 99.8 | 99.9 | 99.9 | 99.9 | 99.6 |
| mnasnet1 0 | 73.46 | 51.5 | 98.6 | 98.9 | 99.8 | 99.9 | 99.9 | 99.9 | 99.9 | 99.9 | 99.9 | 94.8 |
| mobilenet v2 | 71.89 | 22.4 | 99.8 | 99.8 | 99.8 | 99.9 | 99.9 | 99.9 | 99.8 | 99.8 | 99.9 | 92.1 |
| mobilenet v3 large | 74.04 | 23.1 | 81.3 | 94.1 | 95.5 | 96.8 | 98.1 | 98.0 | 98.3 | 99.3 | 99.4 | 88.4 |
| mobilenet v3 small | 67.67 | 55.7 | 68.6 | 97.8 | 99.8 | 99.8 | 99.8 | 99.8 | 99.8 | 99.8 | 99.8 | 92.1 |
| regnet y 16gf | 80.43 | 3.6 | 8.5 | 15.6 | 35.2 | 51.3 | 74.2 | 73.2 | 69.5 | 71.7 | 82.9 | 48.6 |
| regnet y 1 6gf | 77.95 | 21.4 | 36.1 | 50.6 | 43.2 | 54.2 | 63.9 | 67.9 | 69.7 | 75.7 | 78.7 | 56.1 |
| regnet y 32gf | 80.88 | 3.4 | 20.6 | 38.8 | 39.3 | 51.5 | 59.4 | 78.7 | 98.4 | 99.4 | 99.5 | 58.9 |
| regnet y 3 2gf | 78.94 | 10.9 | 16.6 | 35.2 | 39.0 | 52.7 | 70.4 | 79.4 | 81.2 | 93.1 | 85.5 | 56.4 |
| regnet y 400mf | 74.05 | 0.8 | 5.3 | 28.4 | 36.7 | 40.7 | 44.6 | 48.8 | 49.5 | 46.6 | 64.1 | 36.6 |
| regnet y 800mf | 76.43 | 16.3 | 18.7 | 48.7 | 54.8 | 58.8 | 67.5 | 68.4 | 68.6 | 66.8 | 65.8 | 53.5 |
| regnet y 8gf | 80.03 | 12.9 | 33.6 | 49.8 | 51.8 | 85.1 | 83.5 | 83.3 | 86.4 | 87.3 | 91.8 | 66.6 |
| resnet101 | 77.37 | 13.6 | 20.3 | 22.3 | 29.2 | 52.0 | 54.5 | 58.0 | 61.6 | 71.9 | 72.0 | 45.5 |
| resnet152 | 78.32 | 2.1 | 9.1 | 10.1 | 13.7 | 20.5 | 20.8 | 24.7 | 31.1 | 32.0 | 32.4 | 19.7 |
| resnet18 | 69.76 | 10.3 | 22.3 | 27.1 | 41.4 | 41.9 | 43.0 | 40.5 | 43.3 | 45.4 | 70.6 | 38.6 |
| resnet34 | 73.31 | 6.6 | 9.4 | 48.1 | 60.5 | 69.1 | 69.7 | 78.5 | 83.4 | 79.6 | 76.3 | 58.1 |
| resnet50 | 80.39 | 6.7 | 6.6 | 23.0 | 27.0 | 40.4 | 40.3 | 40.2 | 40.8 | 41.1 | 51.9 | 31.8 |
| shufflenet v2 x0 5 | 60.55 | 90.4 | 99.6 | 99.6 | 99.5 | 99.7 | 99.7 | 99.7 | 99.7 | 99.7 | 99.7 | 98.7 |
| shufflenet v2 x1 0 | 69.36 | 93.9 | 99.6 | 99.6 | 99.7 | 99.8 | 99.5 | 99.5 | 99.5 | 99.5 | 99.6 | 99.0 |
| squeezenet1 0 | 58.10 | 12.9 | 18.8 | 23.3 | 25.2 | 37.1 | 49.3 | 49.3 | 53.0 | 53.2 | 77.7 | 40.0 |
| squeezenet1 1 | 58.17 | 10.0 | 44.3 | 72.9 | 72.6 | 83.9 | 87.2 | 89.0 | 92.0 | 91.6 | 92.0 | 73.6 |
| tf efficientnetv2 l | 85.19 | 1.7 | 2.7 | 5.1 | 7.6 | 51.9 | 49.5 | 39.0 | 41.8 | 34.0 | 35.7 | 26.9 |
| tf efficientnetv2 m | 84.55 | 3.0 | 6.5 | 11.1 | 33.2 | 40.7 | 63.7 | 77.3 | 88.3 | 90.5 | 85.9 | 50.0 |
| tf efficientnetv2 s | 83.14 | 3.5 | 7.7 | 20.7 | 29.1 | 52.0 | 47.7 | 77.8 | 77.5 | 88.8 | 90.7 | 49.5 |
| vgg11 bn | 70.38 | 56.3 | 91.3 | 89.5 | 94.3 | 94.7 | 96.9 | 97.5 | 98.2 | 98.8 | 98.7 | 91.6 |
| vgg13 bn | 71.59 | 22.2 | 62.8 | 70.9 | 83.0 | 83.8 | 80.6 | 82.8 | 85.6 | 96.7 | 96.8 | 76.5 |
| vgg16 bn | 73.36 | 13.0 | 53.8 | 58.6 | 60.1 | 69.5 | 76.4 | 81.7 | 82.7 | 88.8 | 88.9 | 67.4 |
| vgg19 bn | 74.22 | 14.1 | 57.8 | 63.1 | 72.2 | 72.2 | 83.9 | 89.4 | 93.0 | 95.2 | 97.4 | 73.8 |
| vit base patch16 224 | 84.53 | 97.2 | 99.5 | 99.8 | 99.8 | 99.9 | 99.8 | 99.8 | 99.8 | 99.8 | 99.8 | 99.5 |
| vit base patch32 224 | 80.71 | 45.1 | 83.4 | 98.4 | 97.6 | 99.7 | 99.8 | 99.8 | 99.8 | 99.8 | 99.8 | 92.3 |
| vit small patch16 224 | 81.39 | 63.4 | 79.5 | 78.9 | 77.8 | 77.7 | 77.7 | 78.2 | 78.4 | 74.7 | 65.5 | 75.2 |
| vit small patch32 224 | 76.00 | 71.2 | 98.8 | 99.0 | 99.7 | 99.6 | 99.6 | 99.7 | 99.7 | 99.7 | 99.7 | 96.7 |
| vit tiny patch16 224 | 75.46 | 99.7 | 99.7 | 99.7 | 99.7 | 99.7 | 99.7 | 99.7 | 99.7 | 99.7 | 99.7 | 99.7 |

Table 10: Full ImageNet results for the pass-free DNL attack. We report the baseline top-1 accuracy, accuracy reduction $AR(k)$, and the average $mAR_{10}$. Results are computed with the one-flip-per-kernel constraint for convolutional models. While most architectures exhibit rapid collapse, some models show more gradual degradation, highlighting architectural differences in vulnerability.

| Model | Base Acc. | AR(1) | AR(2) | AR(3) | AR(4) | AR(5) | AR(6) | AR(7) | AR(8) | AR(9) | AR(10) | Avg. |
|---|---|---|---|---|---|---|---|---|---|---|---|---|
| alexnet | 56.52 | 0.0 | 0.1 | 6.2 | 6.2 | 12.8 | 16.5 | 21.1 | 21.6 | 21.6 | 50.8 | 15.7 |
| convnext base@fb in1k | 83.83 | 3.3 | 17.0 | 6.5 | 8.3 | 11.2 | 22.7 | 28.2 | 35.2 | 39.9 | 60.8 | 23.3 |
| convnext large@fb in1k | 84.29 | 39.6 | 68.5 | 73.4 | 49.8 | 75.3 | 44.7 | 90.7 | 63.2 | 70.1 | 99.6 | 67.5 |
| convnext small@fb in1k | 83.15 | 29.8 | 40.7 | 38.8 | 58.5 | 52.2 | 62.8 | 91.4 | 98.9 | 91.9 | 94.1 | 65.9 |
| convnext tiny@fb in1k | 82.07 | 24.0 | 63.2 | 63.1 | 93.3 | 56.4 | 99.9 | 99.9 | 99.9 | 99.9 | 99.9 | 79.9 |
| efficientnet b0 | 77.69 | 17.6 | 27.7 | 99.6 | 99.6 | 98.8 | 99.4 | 99.5 | 99.9 | 99.8 | 99.9 | 84.2 |
| efficientnet b1 | 80.39 | 1.8 | 1.8 | 2.8 | 4.3 | 99.1 | 99.7 | 12.9 | 16.4 | 73.6 | 66.2 | 37.9 |
| efficientnet b2 | 79.30 | 6.7 | 99.2 | 99.5 | 99.5 | 99.6 | 94.0 | 98.3 | 99.8 | 99.7 | 99.9 | 89.6 |
| efficientnet b3 | 81.49 | 2.1 | 3.2 | 6.9 | 61.9 | 76.5 | 46.7 | 77.9 | 90.7 | 83.6 | 96.6 | 54.6 |
| efficientnet b4 | 82.66 | 2.0 | 30.3 | 90.4 | 40.2 | 75.7 | 99.8 | 99.9 | 9.2 | 10.1 | 99.9 | 55.7 |
| efficientnet b5 | 85.89 | 30.8 | 2.1 | 3.5 | 27.4 | 6.7 | 98.5 | 43.4 | 99.7 | 99.6 | 99.7 | 51.2 |
| efficientnet b6 | 83.84 | 0.9 | 0.4 | 47.4 | 1.4 | 2.8 | 51.3 | 97.6 | 23.2 | 98.3 | 96.7 | 42.0 |
| efficientnet b7 | 84.4 | 3.3 | 2.5 | 1.9 | 7.9 | 6.5 | 8.3 | 11.9 | 25.7 | 27.0 | 27.0 | 12.2 |
| googlenet | 69.78 | 41.0 | 42.9 | 99.0 | 94.9 | 56.1 | 86.6 | 94.5 | 91.0 | 71.5 | 93.6 | 77.1 |
| inception v3 | 77.30 | 9.4 | 55.5 | 76.1 | 73.6 | 92.0 | 77.4 | 94.7 | 66.2 | 71.0 | 97.4 | 71.3 |
| mnasnet0 5 | 67.73 | 99.8 | 99.7 | 99.9 | 99.9 | 99.9 | 99.9 | 99.9 | 99.9 | 99.9 | 99.8 | 99.8 |
| mnasnet1 0 | 73.46 | 10.6 | 69.1 | 87.0 | 99.9 | 99.9 | 99.9 | 99.9 | 99.8 | 99.8 | 99.9 | 86.6 |
| mobilenet v2 | 71.88 | 99.8 | 99.8 | 99.8 | 99.9 | 99.9 | 99.8 | 99.9 | 99.9 | 99.9 | 99.9 | 99.9 |
| mobilenet v3 large | 74.04 | 13.0 | 99.8 | 99.8 | 99.9 | 99.9 | 99.9 | 99.9 | 99.9 | 99.8 | 99.9 | 91.2 |
| mobilenet v3 small | 67.67 | 99.8 | 99.8 | 99.7 | 99.8 | 99.8 | 99.9 | 99.8 | 99.9 | 99.8 | 99.8 | 99.8 |
| regnet y 16gf | 80.42 | 3.6 | 8.5 | 15.6 | 35.1 | 51.3 | 74.2 | 73.2 | 69.5 | 71.7 | 82.9 | 48.6 |
| regnet y 1 6gf | 77.95 | 21.4 | 36.1 | 50.6 | 43.2 | 54.2 | 63.9 | 67.9 | 69.7 | 75.7 | 78.6 | 56.1 |
| regnet y 32gf | 80.88 | 3.4 | 20.6 | 38.8 | 39.3 | 51.5 | 59.4 | 78.7 | 98.4 | 99.4 | 99.5 | 58.9 |
| regnet y 3 2gf | 78.95 | 10.9 | 16.6 | 35.2 | 39.0 | 52.7 | 70.4 | 79.4 | 81.3 | 93.1 | 85.5 | 56.4 |
| regnet y 400mf | 74.05 | 0.8 | 5.3 | 28.4 | 36.7 | 40.7 | 70.2 | 48.8 | 49.5 | 60.1 | 64.1 | 40.5 |
| regnet y 800mf | 76.42 | 16.3 | 18.7 | 48.7 | 54.8 | 58.8 | 67.5 | 68.4 | 92.7 | 84.8 | 87.6 | 59.9 |
| regnet y 8gf | 80.03 | 12.9 | 33.6 | 49.8 | 51.8 | 85.1 | 83.5 | 83.3 | 86.4 | 87.3 | 91.8 | 66.6 |
| resnet101 | 77.37 | 13.6 | 20.3 | 22.3 | 29.2 | 52.0 | 54.5 | 58.0 | 75.5 | 74.8 | 72.0 | 47.2 |
| resnet152 | 78.31 | 2.1 | 9.1 | 10.1 | 13.7 | 20.5 | 25.9 | 24.7 | 31.1 | 31.9 | 32.4 | 20.2 |
| resnet18 | 69.76 | 10.3 | 22.3 | 27.1 | 41.4 | 41.9 | 42.9 | 40.5 | 43.3 | 45.4 | 70.6 | 38.6 |
| resnet34 | 73.32 | 6.6 | 9.4 | 48.1 | 60.5 | 69.1 | 69.7 | 78.5 | 83.4 | 79.6 | 76.3 | 58.1 |
| resnet50 | 80.38 | 99.4 | 99.8 | 99.8 | 99.7 | 99.9 | 99.9 | 99.9 | 99.9 | 99.9 | 99.9 | 99.8 |
| shufflenet v2 x0 5 | 60.55 | 97.6 | 99.3 | 99.7 | 99.7 | 99.8 | 99.7 | 99.8 | 99.9 | 99.8 | 99.9 | 99.5 |
| shufflenet v2 x1 0 | 69.36 | 47.6 | 89.0 | 99.7 | 99.5 | 99.6 | 99.7 | 99.8 | 99.8 | 99.8 | 99.8 | 93.4 |
| squeezenet1 0 | 58.09 | 0.1 | 0.5 | 22.8 | 0.9 | 23.2 | 23.1 | 25.5 | 32.6 | 29.0 | 23.6 | 18.1 |
| squeezenet1 1 | 58.18 | 11.9 | 10.7 | 11.7 | 27.6 | 16.8 | 51.2 | 43.5 | 68.5 | 50.5 | 68.5 | 36.1 |
| tf efficientnetv2 l | 85.83 | 0.9 | 3.9 | 5.6 | 7.3 | 2.1 | 5.9 | 38.4 | 42.0 | 34.1 | 36.2 | 17.6 |
| tf efficientnetv2 m | 84.77 | 1.2 | 3.0 | 28.4 | 46.1 | 52.7 | 64.3 | 76.9 | 89.4 | 89.4 | 86.0 | 53.7 |
| tf efficientnetv2 s | 83.33 | 1.5 | 1.4 | 6.0 | 6.2 | 2.7 | 8.1 | 9.2 | 17.4 | 9.2 | 12.8 | 7.5 |
| vgg11 bn | 70.37 | 56.3 | 91.3 | 89.5 | 94.3 | 94.7 | 96.9 | 97.5 | 98.2 | 98.8 | 98.7 | 91.6 |
| vgg13 bn | 71.59 | 22.2 | 62.8 | 70.9 | 83.0 | 83.8 | 80.6 | 82.8 | 85.6 | 96.7 | 96.8 | 76.5 |
| vgg16 bn | 73.36 | 13.0 | 53.8 | 58.6 | 60.1 | 69.5 | 76.4 | 81.7 | 82.7 | 88.8 | 88.9 | 67.4 |
| vgg19 bn | 74.22 | 14.1 | 57.8 | 63.1 | 72.2 | 72.2 | 83.9 | 89.4 | 93.0 | 95.2 | 97.4 | 73.8 |
| vit base patch16 224 | 85.10 | 17.2 | 84.9 | 29.6 | 90.7 | 97.1 | 87.5 | 99.7 | 94.6 | 97.3 | 99.6 | 79.8 |
| vit base patch32 224 | 80.71 | 45.1 | 83.4 | 98.4 | 97.6 | 99.7 | 99.8 | 99.8 | 99.8 | 99.8 | 99.8 | 92.3 |
| vit small patch16 224 | 81.39 | 63.4 | 79.5 | 78.9 | 77.8 | 77.7 | 77.7 | 78.1 | 78.4 | 74.7 | 65.6 | 75.2 |
| vit small patch32 224 | 75.99 | 71.2 | 98.8 | 99.0 | 99.7 | 99.7 | 99.6 | 99.7 | 99.7 | 99.7 | 99.7 | 96.7 |
| vit tiny patch16 224 | 75.47 | 99.7 | 99.7 | 99.7 | 99.7 | 99.7 | 99.7 | 99.7 | 99.7 | 99.7 | 99.7 | 99.7 |

Table 11: Full ImageNet results for the single-pass 1P-DNL attack. Compared to the pass-free variant, incorporating gradient information from a single forward/backward pass significantly increases attack strength, especially at low flip budgets. Many models reach near-complete collapse with only a few sign-bit flips.

