# OpenReview forum: "Maximal Brain Damage Without Data or Optimization: Disrupting Neural Networks via Sign-Bit Flips"
_TMLR — Accepted by TMLR_

### Review · Reviewer_zVfB · 2025-12-10

**Summary Of Contributions:**

This paper introduces a novel adversarial attack method termed Deep Neural Lesion (DNL), in which flipping a small number of sign‑bits in a trained neural network’s parameters causes catastrophic performance degradation. The attack is data‑free and requires no optimization or access to training or inference data. As long as the attacker obtains the model parameters, they can destroy the model’s functionality by flipping only a few sign bits. The authors also propose corresponding defense mechanisms to mitigate this vulnerability.

**Audience:**

Yes

**Audience Explanation:**

Yes. The work exposes a concrete, under‑explored vulnerability, and its findings have the potential to influence how researchers think about model robustness and deployment safety.

**Broader Impact Concerns:**

None ethical concerns

**Claims And Evidence:**

Yes

**Claims Explanation:**

Yes. I believe the submission provides strong and convincing evidence for its core claim under the narrow, evaluated regime: namely, that floating‑point classification models can be catastrophically disrupted by flipping just a few sign bits. The methodology is well-explained and experiments support the claim clearly. However, the evidence is not sufficient to support broader claims about general neural‑network vulnerability: the lack of evaluation across tasks, architectures, and quantized models. For these broader claims to be convincing, the authors would need additional experiments and deeper analysis.

**Requested Changes:**

1. Although the authors evaluate the proposed method on a variety of network architectures (e.g., ResNet, ViT, CNN), the evaluation is restricted to classification tasks. There is a lack of experiments on other tasks such as detection, segmentation, or generation. I recommend extending the evaluation to more tasks.
2. Some of the baseline comparison methods appear outdated. The authors should add comparisons with more recent methods.
3. In modern practice many models use transformer-based backbones beyond ViT (e.g., architectures such as Swin‑Transformer). The authors only discuss ViT; it would strengthen the paper to include results on whether DNL is effective on such representative alternative architectures.
4. The authors should more thoroughly analyze DNL’s limitations.
5. All experiments appear to use 32-bit floating‑point weights. Yet, many deployed models use weight quantization. The authors should discuss whether the proposed method works on quantized models (e.g., INT8 or other reduced-precision formats).

---

> ### Author Response · Authors · 2026-01-05
> **Official comment**
>
> Thank you for your positive feedback,
>
> >*”There is a lack of experiments on other tasks such as detection, segmentation, or generation. I recommend extending the evaluation to more tasks.”*
>
> We agree, and we extended the evaluation beyond vision classification.
>
> **Object detection + instance segmentation (COCO2017).** We evaluated our standard DNL attack on:
> 1) Mask R-CNN (FPN) with ResNet-50 and ResNet-101 backbones (torchvision weights)
> 2) YOLOv8-seg (Ultralytics)
>
> We report AP@[0.50:0.95] and AP@0.50 for boxes (bbox) and masks (segm). Even k=1–2 sign flips are sufficient to collapse performance.
>
> | Model \ Backbone | Metric | Baseline | Post-attack (K=1) | AR(1) | Post-attack (K=2) | AR(2) |
> |--|--|--:|---:|--:|--:|--:|
> | ResNet-50 | bbox_AP | 0.38 | 0.01 | 97.36 | 0 | 100 |
> |  | bbox_AP50 | 0.59 | 0.03 | 94.93 | 0 | 100 |
> |  | segm_AP | 0.35 | 0 | 100 | 0 | 100 |
> |  | segm_AP50 | 0.56 | 0.01 | 98.21 | 0 | 100 |
> | ResNet-101 | bbox_AP | 0.4 | 0.01 | 97.51 | 0.01 | 97.51 |
> |  | bbox_AP50 | 0.61 | 0.03 | 95.12 | 0.02 | 96.75 |
> |  | segm_AP | 0.36 | 0 | 100 | 0 | 100 |
> |  | segm_AP50 | 0.58 | 0.01 | 98.28 | 0 | 100 |
> | YOLO-V8-SEG | bbox_AP | 0.33 | 0.05 | 83.66 | 0.05 | 86.33 |
> |  | bbox_AP50 | 0.47 | 0.08 | 82.01 | 0.07 | 84.92 |
> |  | segm_AP | 0.05 | 0.01 | 77.8 | 0.01 | 80.51 |
> |  | segm_AP50 | 0.16 | 0.04 | 77.73 | 0.03 | 81.28 |
>
> **NLP (transformers).** We also ran experiments in the NLP domain with BERT, RoBERTa, and DistilBERT. Interestingly, unlike the vision models we tested, the most vulnerable bit we observed in these NLP models is often in the exponent.
> Below we report AR(10) for our standard DNL attack on each model across three tasks: SST2, QNLI, and MRPC.
>
> |Model|Task||AR(10)|
> |-|-|:-:|--:|
> |RoBERTa-base | sst2 | &nbsp; | 63.94 |
> | | qnli || 49.99 |
> | | mrpc || 65.32 |
> |BERT-base | sst2 | &nbsp; | 50.11 |
> | | qnli | &nbsp; | 49.03 |
> | | mrpc | &nbsp; | 63.97 |
> |DistilBERT-base | sst2 | &nbsp; | 49.04 |
> | | qnli | &nbsp; | 46.49 |
> | | mrpc | &nbsp; | 62.72 |
>
> We will include these additional results in the revision.
>
> >*”All experiments appear to use 32-bit floating‑point weights. Yet, many deployed models use weight quantization. The authors should discuss whether the proposed method works on quantized models”*
>
> Table 1 in Section 5 already reports ImageNet INT8 quantized models, and shows that DNL and 1P-DNL achieve 99%+ accuracy reduction with less than 10 flips. Table 6 additionally reports DNL results on a binary ResNet-18. We agree this should be highlighted more clearly in our revision.
>
> >*”The authors only discuss ViT; it would strengthen the paper to include results on whether DNL is effective on such representative alternative architectures”*
>
> Beyond ViT and the aforementioned NLP transformer models, we also ran DNL and 1P-DNL on a couple of other transformer architectures we found interesting:
>
> |Model|DNL AR(10)|1P-DNL AR(10)|
> |-|-:|-:|
> |beit3 base (patch16 224) [1]|82.93|85.2|
> |PVT [2]|55.19|70.85|
>
>
> >*”Some of the baseline comparison methods appear outdated”*
>
>
> Our threat model is intentionally restrictive (no real data; and limited / no forward/backward passes). Under these constraints, we are not aware of newer methods that provide a truly apples-to-apples comparison. The baselines in Section 5 are still widely used references in the weight-bit attack literature, and we include them to contextualize how our low-budget attack compares in terms of success.
>
> If you have a specific recent method in mind that matches these constraints, we would be happy to consider evaluating it.
>
> >*”The authors should more thoroughly analyze DNL’s limitations.”*
>
> We agree and will revise the paper with a more thorough analysis. We believe the main limitation of DNL is its dependence on access to model parameters. Below is a paragraph we intend to add to the paper:
>
>
> *Limitations: DNL assumes an adversary can directly modify model parameters by flipping a small number of bits in the stored weights. The practical feasibility of this assumption depends on whether the attacker can reach and change specific parameter locations, which may be only partially possible in some deployments (for example, if the model is split across components, stored in separate shards, or only a subset of weights is writable or addressable). In such access-constrained settings, a naive global search over all weights is not applicable and the attack’s impact may be reduced. At the same time, the empirical patterns underlying our pass-free variant, namely that damage is often concentrated in a small subset of layers and high-magnitude weights, motivate studying more targeted attacks (and corresponding defenses) under partial-access threat models.
> ​​The acess limitation is consistent with the threat model described in our problem setup.*
>
> [1] Image as a Foreign Language: BEiT Pretraining for All Vision and Vision-Language Tasks
>
> [2] Pyramid Vision Transformer: A Versatile Backbone for Dense Prediction without Convolutions

---

> > ### Author Response · Authors · 2026-01-16
> > **Experiments on LLMs**
> >
> > Dear zVfB,
> >
> > Since our last response, continuing our discussion on NLP models, we also evaluated DNL and 1P-DNL on several LLMs. Remarkably, we found that k=1 (a single bit flip) is enough to completely disrupt generation: the model begins producing **nonsensical output and loses basic linguistic structure**.
> >
> > To make this quantitative, we tested three models: Qwen3-4B-Thinking, Qwen3-30B-A3B-Thinking, and Llama-3.1-Nemotron-Nano-8B-v1, and evaluated them on a stratified sample from the MATH500 benchmark (see [1] for technical details).
> >
> > All three models collapsed to 0% accuracy at k=1, as shown below:
> >
> > | Model | Baseline accuracy | Accuracy at DNL (k=1) | Accuracy at 1P-DNL (k=1) |
> > |---|---:|---:|---:|
> > | Qwen3-30B-A3B-Thinking | 98% | 0% | 0% |
> > | Qwen3-4B-Thinking | 100% | 0% | 0% |
> > | Llama-3.1-Nemotron-Nano-8B-v1 | 94% | 0% | 0% |
> >
> > Given the loss of coherent generation, the drop to 0% is unsurprising and would most likely repeat for any other benchmark.
> >
> > Importantly, in these LLMs the single-bit failure is not confined to a specific module. For example, in Llama-3.1-Nemotron-Nano-8B-v1 the flipped bit landed in an attention layer, but even when we constrain the search to FFN weights only, we still observe the same collapse to 0% at k=1.
> >
> > Notably, Qwen3-30B-A3B-Thinking is a mixture-of-experts model, and the flipped bit belonged to a single expert. Even though that expert is not active at every token, the generation was out of structure throughout the entire sequence. Our working hypothesis is that this is a direct consequence of autoregressive causality: it can be enough for the perturbation to corrupt part of the prompt’s latent representations, after which later tokens attend to these corrupted states and the error propagates and compounds across the context. This compounding effect is largely absent in standard classifiers (such as the image classifiers we reported in the paper), where each inference is independent, which may help explain why k=1 can be so destructive in LLM generation.
> >
> > [1] We evaluated using the EleutherAI lm-evaluation-harness (https://github.com/EleutherAI/lm-evaluation-harness), using “minerva_math” with its SymPy-based equivalence check (rather than strict exact match). We used a stratified sample by difficulty level (5 questions per level; 50 total) from MATH500.

---

### Review · Reviewer_Jwin · 2025-12-15

**Summary Of Contributions:**

This paper identifies a vulnerability in deep neural networks at the parameter bit level, named Deep Neural Lesion (DNL). Specifically, DNL shows that flipping the sign bit of a very small number of FP32 weights can cause catastrophic degradation in model performance. The proposed DNL attack is data-free, optimization-free, and pass-free. The paper further introduces 1P-DNL, a stronger variant that allows a single forward and backward pass using random inputs to enhance impact. Extensive experiments on 60 classifiers demonstrate that even one or two sign-bit flips can significantly reduce accuracy.

Strengths:
1. The proposed DNL attack is a critical threat to deep neural networks. The attack is conceptually simple, computationally cheap, and easy to implement.
2. The visualization (e.g., Fig. 1) is good. These visualizations clearly show how the DNL attack affected the model’s performance on vision tasks.
3. Empirical results show the DNL attack is effective and these results are consistent across multiple architectures and datasets.
4. The paper is well written and well organized.

Weaknesses:
1. Experiments are limited to vision classification models.
2. The computation of the hybrid importance score is computational expensive and relies on specific approximations.

**Additional Comments:**

Overall, this is a well-written and insightful paper that exposes an important and critical vulnerability in deep neural networks. While additional experimental validation would further strengthen the claims, the core insight that model accuracy can hinge on only a handful of sign bits is  likely to stimulate future research on secure model deployment.

**Audience:**

Yes

**Audience Explanation:**

The paper is highly relevant to researchers interested in robustness and security of deep neural networks. The simplicity and severity of the proposed DNL attack make the findings noteworthy for a broad ML community.

**Broader Impact Concerns:**

The simplicity and low implementation cost of the proposed attacks significantly lower the barrier for malicious misuse. While the paper includes potential defense strategies, a comprehensive discussion of plausible misuse cases would help readers better understand the ethical implications of the work.

**Claims And Evidence:**

Yes

**Claims Explanation:**

The central claim that a tiny number of sign-bit flips can catastrophically degrade accuracy, is demonstrated through empirical results in Table 1. Furthermore, the effectiveness of the proposed DNL attack is demonstrated by the comparison with other attack strategies such as random bit flips in Figure 2.

**Requested Changes:**

Critical:
1. Include more non-vision models (e.g., a Transformer or NLP task) to test whether the vulnerability generalizes across diverse model architectures and tasks.
2. Include additional experiments comparing with other model pruning strategies, and discuss the differences in parameter selection between the DNL attack and model pruning tasks.

Non-critical:
3. The practical attack feasibility is discussed largely at a conceptual level (e.g., references to DMA). It could be better to show quantitative evidence of attack success rates in real deployment settings.

---

> ### Author Response · Authors · 2026-01-05
>
> Thank you for your constructive feedback,
>
> >*”Include more non-vision models (e.g., a Transformer or NLP task) to test whether the vulnerability generalizes across diverse model architectures and tasks.”*
>
> We extended our evaluation beyond vision classification.
>
> We ran experiments in the NLP domain with BERT, RoBERTa, and DistilBERT. Interestingly, unlike the vision models we tested, the most vulnerable bit we observed in these NLP models is often in the exponent.
> Below we report AR(10) for our standard DNL attack on each model across three tasks: SST2, QNLI, and MRPC.
>
> | Model | Task | &nbsp;&nbsp;&nbsp; | AR(10) |
> |---|---|:--:|---:|
> | RoBERTa-base | sst2 | &nbsp; | 63.94 |
> |  | qnli | &nbsp; | 49.99 |
> |  | mrpc | &nbsp; | 65.32 |
> | BERT-base | sst2 | &nbsp; | 50.11 |
> |  | qnli | &nbsp; | 49.03 |
> |  | mrpc | &nbsp; | 63.97 |
> | DistilBERT-base | sst2 | &nbsp; | 49.04 |
> |  | qnli | &nbsp; | 46.49 |
> |  | mrpc | &nbsp; | 62.72 |
>
> We will include these additional results in the revision.
>
> In addition, we extended beyond vision classification to object detection and instance segmentation on COCO2017 (Mask R-CNN with ResNet-50/101 backbones and YOLOv8-seg), where k=1–2 sign flips severely degrade AP.
>
> | Model \ Backbone | Metric | Baseline | Post-attack (K=1) | AR(1) | Post-attack (K=2) | AR(2) |
> |---|---|---:|---:|---:|---:|---:|
> | ResNet-50 | bbox_AP | 0.38 | 0.01 | 97.36 | 0 | 100 |
> |  | bbox_AP50 | 0.59 | 0.03 | 94.93 | 0 | 100 |
> |  | segm_AP | 0.35 | 0 | 100 | 0 | 100 |
> |  | segm_AP50 | 0.56 | 0.01 | 98.21 | 0 | 100 |
> | ResNet-101 | bbox_AP | 0.4 | 0.01 | 97.51 | 0.01 | 97.51 |
> |  | bbox_AP50 | 0.61 | 0.03 | 95.12 | 0.02 | 96.75 |
> |  | segm_AP | 0.36 | 0 | 100 | 0 | 100 |
> |  | segm_AP50 | 0.58 | 0.01 | 98.28 | 0 | 100 |
> | YOLO-V8-SEG | bbox_AP | 0.33 | 0.05 | 83.66 | 0.05 | 86.33 |
> |  | bbox_AP50 | 0.47 | 0.08 | 82.01 | 0.07 | 84.92 |
> |  | segm_AP | 0.05 | 0.01 | 77.8 | 0.01 | 80.51 |
> |  | segm_AP50 | 0.16 | 0.04 | 77.73 | 0.03 | 81.28 |
>
>
> >*”The computation of the hybrid importance score is computational expensive and relies on specific approximations.”*
>
> While 1P-DNL is more expensive than DNL, it requires only a single forward/backward pass on a random (Gaussian) input. The approximations we use are standard in second-order pruning: a diagonal Hessian approximation (ignoring off-diagonal terms) and a Gauss-Newton style approximation for the diagonal term. More importantly, it works well and improves over DNL. DNL itself is fully pass-free and remains the primary attack under the strict threat model. 1P-DNL is included to quantify the additional damage achievable with a single pass. As summarized in Table 4, competing methods typically rely on extensive iterative optimization (and often real data), whereas 1P-DNL uses a single pass with random input and no iterative search.
>
>
> > *”The practical attack feasibility is discussed largely at a conceptual level (e.g., references to DMA). It could be better to show quantitative evidence of attack success rates in real deployment settings.”*
>
> We feel that, beyond outlining plausible attack vectors, demonstrating an end-to-end deployment-specific exploit is outside the scope of this work, which is centered on exposing and quantifying a *model-level* vulnerability. In practice, how an attacker could realize parameter tampering depends heavily on the particular hardware/software stack and operational environment, and new (possibly simpler) mechanisms could emerge in future deployments that are orthogonal to any specific exploit we would demonstrate today.
>
> This is also common in closely related security literatures: seminal non-physical adversarial example works (e.g., FGSM, C&W, PGD) typically assume the attacker can supply or modify the model input x, without demonstrating the concrete end-to-end method by which that capability is obtained in a deployed system. Our paper makes an analogous abstraction: given write access to stored parameters, we show that flipping a very small number of bits can reliably cause catastrophic degradation.
>
> >*"Include additional experiments comparing with other model pruning strategies, and discuss the differences in parameter selection between the DNL attack and model pruning tasks.”*
>
> We already include a pruning-oriented comparison in Appendix D, where we evaluate strategies from the pruning literature, such as magnitude-based, SynFlow, GraSP and OBD-style criteria. We compare them to our hybrid score.
>
> Are there any other methods you find relevant to add? If so, we will try to include them in the revision.

---

> > ### Author Response · Authors · 2026-01-16
> > **Experiments on LLMs**
> >
> > Dear Jwin,
> >
> > Since our last response, continuing our discussion on NLP models, we also evaluated DNL and 1P-DNL on several LLMs. Remarkably, we found that k=1 (a single bit flip) is enough to completely disrupt generation: the model begins producing **nonsensical output and loses basic linguistic structure**.
> >
> > To make this quantitative, we tested three models: Qwen3-4B-Thinking, Qwen3-30B-A3B-Thinking, and Llama-3.1-Nemotron-Nano-8B-v1, and evaluated them on a stratified sample from the MATH500 benchmark (see [1] for technical details).
> >
> > All three models collapsed to 0% accuracy at k=1, as shown below:
> >
> > | Model | Baseline accuracy | Accuracy at DNL (k=1) | Accuracy at 1P-DNL (k=1) |
> > |---|---:|---:|---:|
> > | Qwen3-30B-A3B-Thinking | 98% | 0% | 0% |
> > | Qwen3-4B-Thinking | 100% | 0% | 0% |
> > | Llama-3.1-Nemotron-Nano-8B-v1 | 94% | 0% | 0% |
> >
> > Given the loss of coherent generation, the drop to 0% is unsurprising and would most likely repeat for any other benchmark.
> >
> > Importantly, in these LLMs the single-bit failure is not confined to a specific module. For example, in Llama-3.1-Nemotron-Nano-8B-v1 the flipped bit landed in an attention layer, but even when we constrain the search to FFN weights only, we still observe the same collapse to 0% at k=1.
> >
> > Notably, Qwen3-30B-A3B-Thinking is a mixture-of-experts model, and the flipped bit belonged to a single expert. Even though that expert is not active at every token, the generation was out of structure throughout the entire sequence. Our working hypothesis is that this is a direct consequence of autoregressive causality: it can be enough for the perturbation to corrupt part of the prompt’s latent representations, after which later tokens attend to these corrupted states and the error propagates and compounds across the context. This compounding effect is largely absent in standard classifiers (such as the image classifiers we reported in the paper), where each inference is independent, which may help explain why k=1 can be so destructive in LLM generation.
> >
> > [1] We evaluated using the EleutherAI lm-evaluation-harness (https://github.com/EleutherAI/lm-evaluation-harness), using “minerva_math” with its SymPy-based equivalence check (rather than strict exact match). We used a stratified sample by difficulty level (5 questions per level; 50 total) from MATH500.

---

### Review · Reviewer_N7Um · 2025-12-30

**Summary Of Contributions:**

This paper introduces a data-free, optimization-free attack that flips a small number of sign bits in neural network weights to cause catastrophic accuracy drops. Operates under an extremely restrictive threat model: no access to training data and no forward/backward passes.

**Additional Comments:**

Overall, I think the contribution of this paper is not novel enought over existing works on neural network compression - an active research area back to 2015.

**Audience:**

Yes

**Audience Explanation:**

**Security and Robustness Researchers:** The paper exposes a critical vulnerability in neural networks that can be exploited without data or optimization, which is highly relevant for those studying adversarial robustness and model security.

**Systems and Deployment Engineers:** Lightweight attacks that require minimal computation and no data are concerning for real-world deployments (e.g., autonomous driving, medical AI), making this work important for practitioners.

**Model Compression and Pruning Community:** The connection between sign-bit flips and pruning heuristics (magnitude-based selection, Hessian approximations) may interest researchers working on network sparsity and weight importance.

**Defense Strategy Designers:** The proposed selective protection mechanism offers insights into efficient defenses, which is valuable for those designing robust architectures.

However, I feel the proposed work is not very novel.

**Claims And Evidence:**

No

**Claims Explanation:**

**Empirical Evidence:** The paper provides extensive experiments on 60 classifiers (including 48 ImageNet models) and multiple datasets. Results consistently show catastrophic accuracy drops (often >99%) with very few sign-bit flips. This strongly supports the core claim that sign-bit flips can severely disrupt DNNs.


**Clarity of Evidence:** Figures and tables (e.g., Table 1, Figure 2) illustrate the impact of attacks across architectures. However: Variance or confidence intervals are missing, making robustness unclear. Results are limited to vision tasks; claims of generality across domains (e.g., NLP) are not empirically validated.

**Accuracy of Claims:** The claim that attacks are “feasible in real-world scenarios” is not convincingly supported: The paper lists hardware/software exploits (Rowhammer, DMA, rootkits) but does not quantify success rates or practical constraints. No demonstration of actual attack implementation under realistic conditions.

**Theoretical Justification:** The Taylor expansion argument for magnitude-based selection is plausible but relies on strong assumptions (diagonal Hessian, constant curvature) without empirical validation. The Lipschitz-based reasoning for early-layer vulnerability is asserted but not rigorously proven.

**Defense Claims:** The proposed selective defense shows promising results in controlled experiments, but scalability and cost trade-offs are not analyzed.

**Requested Changes:**

**Clarify Practical Feasibility**
1. Provide quantitative analysis or experimental evidence showing that precise sign-bit flips are achievable under realistic hardware/software attack scenarios.
2. Discuss limitations of Rowhammer, DMA, and firmware exploits in targeting specific bits in large models.



**Strengthen Theoretical Justification**
1. Validate assumptions behind the Taylor expansion (diagonal Hessian, constant curvature) with empirical evidence.
2. Quantify the Lipschitz-based amplification argument for early-layer vulnerability rather than relying on analogy.


**Broaden Evaluation Scope**
1. Include experiments on non-vision tasks (e.g., NLP, speech) to support claims of generality.
2. Report variance or confidence intervals for accuracy reduction to demonstrate robustness across trials.

---

> ### Author Response · Authors · 2026-01-05
>
> Thank you for your detailed feedback,
>
> >*”Include experiments on non-vision tasks (e.g., NLP, speech) to support claims of generality.”*
>
> We extended our evaluation beyond vision classification.
>
> We ran experiments in the NLP domain with BERT, RoBERTa, and DistilBERT. Interestingly, unlike the vision models we tested, the most vulnerable bit we observed in these NLP models is often in the exponent.
> Below we report AR(10) for our standard DNL attack on each model across three tasks: SST2, QNLI, and MRPC.
>
> | Model | Task | &nbsp;&nbsp;&nbsp; | AR(10) |
> |---|---|:--:|---:|
> | RoBERTa-base | sst2 | &nbsp; | 63.94 |
> |  | qnli | &nbsp; | 49.99 |
> |  | mrpc | &nbsp; | 65.32 |
> | BERT-base | sst2 | &nbsp; | 50.11 |
> |  | qnli | &nbsp; | 49.03 |
> |  | mrpc | &nbsp; | 63.97 |
> | DistilBERT-base | sst2 | &nbsp; | 49.04 |
> |  | qnli | &nbsp; | 46.49 |
> |  | mrpc | &nbsp; | 62.72 |
>
> We will include these additional results in the revision.
>
> In addition, we extended beyond vision classification to object detection and instance segmentation on COCO2017 (Mask R-CNN with ResNet-50/101 backbones and YOLOv8-seg), where k=1–2 sign flips severely degrade AP.
>
> | Model \ Backbone | Metric | Baseline | Post-attack (K=1) | AR(1) | Post-attack (K=2) | AR(2) |
> |---|---|---:|---:|---:|---:|---:|
> | ResNet-50 | bbox_AP | 0.38 | 0.01 | 97.36 | 0 | 100 |
> |  | bbox_AP50 | 0.59 | 0.03 | 94.93 | 0 | 100 |
> |  | segm_AP | 0.35 | 0 | 100 | 0 | 100 |
> |  | segm_AP50 | 0.56 | 0.01 | 98.21 | 0 | 100 |
> | ResNet-101 | bbox_AP | 0.4 | 0.01 | 97.51 | 0.01 | 97.51 |
> |  | bbox_AP50 | 0.61 | 0.03 | 95.12 | 0.02 | 96.75 |
> |  | segm_AP | 0.36 | 0 | 100 | 0 | 100 |
> |  | segm_AP50 | 0.58 | 0.01 | 98.28 | 0 | 100 |
> | YOLO-V8-SEG | bbox_AP | 0.33 | 0.05 | 83.66 | 0.05 | 86.33 |
> |  | bbox_AP50 | 0.47 | 0.08 | 82.01 | 0.07 | 84.92 |
> |  | segm_AP | 0.05 | 0.01 | 77.8 | 0.01 | 80.51 |
> |  | segm_AP50 | 0.16 | 0.04 | 77.73 | 0.03 | 81.28 |
>
> >*”Report variance or confidence intervals for accuracy reduction to demonstrate robustness across trials.”*
>
> DNL is fully **deterministic**: for a fixed model, layer budget$L$, and flip budget $k$, it selects the same weights and produces the same $AR(k)$ every run. Therefore, there is **no** trial-to-trial variance to report for DNL.
> 1P-DNL uses **a single random Gaussian input** to compute one forward/backward pass and derive the scoring function. This is the only source of stochasticity. In practice, its effect is negligible: evaluating a representative subset of architectures (ConvNeXt-B, RegNetY-400MF, ResNet-50, EfficientNet-B0, and ViT-B/16) over 10 random seeds yields a standard deviation of **0.02**, which is orders of magnitude smaller than the induced accuracy drops. We will include this result in the revision to demonstrate our method’s robustness.
> Given this determinism, the only meaningful variability in our study is across **50 models** and across **5 datasets** rather than across *repeated trials*. We therefore report distributions over architectures (Figures 2, 3, 4, 6, and 11) and across datasets (Figures 7 and 8), which more accurately capture the robustness and generality of the attack.
>
> >*”Provide quantitative analysis or experimental evidence showing that precise sign-bit flips are achievable under realistic hardware/software attack scenarios.”*
> >*”Discuss limitations of Rowhammer, DMA,...”*
>
> We feel that, beyond outlining plausible attack vectors, demonstrating an end-to-end deployment-specific exploit is outside the scope of this work, which is centered on exposing and quantifying a *model-level* vulnerability. In practice, how an attacker could realize parameter tampering depends heavily on the particular hardware/software stack and operational environment, and new (possibly simpler) mechanisms could emerge in future deployments that are orthogonal to any specific exploit we would demonstrate today.
>
> This is also common in closely related security literatures: seminal non-physical adversarial example works (e.g., FGSM, C&W, PGD) typically assume the attacker can supply or modify the model input x, without demonstrating the concrete end-to-end method by which that capability is obtained in a deployed system. Our paper makes an analogous abstraction: given write access to stored parameters, we show that flipping a very small number of bits can reliably cause catastrophic degradation.
>
> That said, we do discuss existing exploits in the problem setup at "Mechanisms enabling parameter bit flips".

---

> ### Author Response · Authors · 2026-01-05
> **Additional Response**
>
> >Taylor expansion assumptions (diagonal Hessian / constant curvature)
>
> Our Taylor-style motivation mirrors standard *second-order pruning* analyses, where exact Hessian-based selection is intractable and diagonal / approximate-curvature proxies are routinely adopted [1,2,3,4,5,6,7].
> Observing the case of a **single sign flip**, the quadratic term depends only on $H_{ii}$ **and** *off-diagonal coupling vanishes*. For $k>1$, neglecting cross-terms is precisely the common diagonal approximation used throughout pruning/continual-learning curvature methods [1,2,6,8].
>
> Moreover, first-/second-order saliencies such as Taylor/gradient x weight and Gauss–Newton/Fisher-diagonal surrogates are widely used in one-shot pruning to rank “important” parameters or even channels, despite relying on the same local quadratic viewpoint [3,4,5,6,7].
>
> To directly address your request, we will add citations that advocate for our commonly used assumptions.
>
>
> >*”Quantifying Lipschitz-based early-layer amplification”*
>
> We would like to clarify that this discussion is intended as **motivation**, not as a formal explanation or proof of why early-layer targeting is optimal. The main justification for the heuristic in this paper is empirical: restricting candidate flips to early layers substantially increases attack impact (as demonstrated in Figure 2).
>
> The Lipschitz composition bound is included only as one plausible interpretation of why early-layer perturbations can be more damaging. In simple terms, a perturbation injected at depth $\ell$ can, in the worst case, be amplified by up to $ \prod_{j>\ell} L_j $ through subsequent layers. This viewpoint is widely used in robustness and generalization analyses of deep networks, as well as in work that explicitly constrains or regularizes network Lipschitz constants [9,10,11,12,13].
>
> In the revision, we can rephrase this section to make the status of the argument explicit (motivation only), and optionally move it to an appendix or remove it entirely if it distracts from the main empirical point.
>
>
> >*”Overall, I think the contribution of this paper is not novel enought over existing works on neural network compression - an active research area back to 2015.”*
>
> Respectfully, we disagree. While pruning/compression methods often study how to identify *least important* parameters so accuracy is preserved after removing many of them, our goal is the opposite: to identify a tiny set of *most damaging* parameters such that flipping only a few sign bits (often 1–2, and up to 10) catastrophically breaks the model. Moreover, pruning is typically performed by an engineer with full control over the model and the training/deployment pipeline, whereas our method operates under a restrictive adversarial threat model (data-free and pass-free). This is a completely different regime and objective than compression.
>
> References:
>
> [1] LeCun, Y., Denker, J. S., & Solla, S. A.
>  Optimal Brain Damage.
>  NeurIPS, 1989.
>
> [2] Hassibi, B., & Stork, D. G.
>  Second Order Derivatives for Network Pruning: Optimal Brain Surgeon.
>  NeurIPS, 1993.
>
> [3] Molchanov, P., Tyree, S., Karras, T., Aila, T., & Kautz, J.
>  Pruning Convolutional Neural Networks for Resource Efficient Inference.
>  ICLR, 2017.
>
> [4] Lee, N., Ajanthan, T., & Torr, P. H. S.
>  SNIP: Single-shot Network Pruning based on Connection Sensitivity.
>  ICLR, 2019.
>
> [5] Wang, C., Zhang, G., & Grosse, R.
>  Picking Winning Tickets Before Training by Preserving Gradient Flow.
>  ICLR, 2020. (GraSP)
>
> [6] Tanaka, H., Kunin, D., Yamins, D. L. K., & Ganguli, S.
>  Pruning Neural Networks without Any Data by Iterative Synaptic Flow.
>  NeurIPS, 2020. (SynFlow)
>
> [7] Frankle, J., & Carbin, M.
>  The Lottery Ticket Hypothesis: Finding Sparse, Trainable Neural Networks.
>  ICLR, 2019.
>
> [8] Kirkpatrick, J. et al.
>  Overcoming Catastrophic Forgetting in Neural Networks.
>  PNAS, 2017. (diagonal curvature approximations in continual learning)
>
> [9] Burago, D., Burago, Y., & Ivanov, S.
>  A Course in Metric Geometry.
>  AMS, 2001.
>
> [10] Sokolić, J., Giryes, R., Sapiro, G., & Rodrigues, M. R. D.
>  Robust Large Margin Deep Neural Networks.
>  IEEE Transactions on Signal Processing, 2017.
>
> [11] Virmaux, A., & Scaman, K.
>  Lipschitz Regularity of Deep Neural Networks: Analysis and Efficient Estimation.
>  NeurIPS, 2018.
>
> [12] Miyato, T., Kataoka, T., Koyama, M., & Yoshida, Y.
>  Spectral Normalization for Generative Adversarial Networks.
>  ICLR, 2018.
>
> [13] Gouk, H., Frank, E., Pfahringer, B., & Cree, M.
>  Regularisation of Neural Networks by Enforcing Lipschitz Continuity.
>  Machine Learning, 2021.

---

> > ### Author Response · Authors · 2026-01-16
> > **Experiments on LLMs**
> >
> > Dear N7Um,
> >
> > Since our last response, continuing our discussion on NLP models, we also evaluated DNL and 1P-DNL on several LLMs. Remarkably, we found that k=1 (a single bit flip) is enough to completely disrupt generation: the model begins producing **nonsensical output and loses basic linguistic structure**.
> >
> > To make this quantitative, we tested three models: Qwen3-4B-Thinking, Qwen3-30B-A3B-Thinking, and Llama-3.1-Nemotron-Nano-8B-v1, and evaluated them on a stratified sample from the MATH500 benchmark (see [1] for technical details).
> >
> > All three models collapsed to 0% accuracy at k=1, as shown below:
> >
> > | Model | Baseline accuracy | Accuracy at DNL (k=1) | Accuracy at 1P-DNL (k=1) |
> > |---|---:|---:|---:|
> > | Qwen3-30B-A3B-Thinking | 98% | 0% | 0% |
> > | Qwen3-4B-Thinking | 100% | 0% | 0% |
> > | Llama-3.1-Nemotron-Nano-8B-v1 | 94% | 0% | 0% |
> >
> > Given the loss of coherent generation, the drop to 0% is unsurprising and would most likely repeat for any other benchmark.
> >
> > Importantly, in these LLMs the single-bit failure is not confined to a specific module. For example, in Llama-3.1-Nemotron-Nano-8B-v1 the flipped bit landed in an attention layer, but even when we constrain the search to FFN weights only, we still observe the same collapse to 0% at k=1.
> >
> > Notably, Qwen3-30B-A3B-Thinking is a mixture-of-experts model, and the flipped bit belonged to a single expert. Even though that expert is not active at every token, the generation was out of structure throughout the entire sequence. Our working hypothesis is that this is a direct consequence of autoregressive causality: it can be enough for the perturbation to corrupt part of the prompt’s latent representations, after which later tokens attend to these corrupted states and the error propagates and compounds across the context. This compounding effect is largely absent in standard classifiers (such as the image classifiers we reported in the paper), where each inference is independent, which may help explain why k=1 can be so destructive in LLM generation.
> >
> > [1] We evaluated using the EleutherAI lm-evaluation-harness (https://github.com/EleutherAI/lm-evaluation-harness), using “minerva_math” with its SymPy-based equivalence check (rather than strict exact match). We used a stratified sample by difficulty level (5 questions per level; 50 total) from MATH500.

---

### Review · Reviewer_CEZm · 2026-01-04

**Summary Of Contributions:**

The authors introduce Deep Neural Lesion (DNL), a lightweight, data-free adversarial attack that catastrophically disrupts deep neural networks by flipping a minimal number of sign bits in critical parameters. The paper proposes two variants: a "Pass-free" method relying solely on weight magnitude heuristics, and a "1-Pass" method utilizing a hybrid gradient-magnitude score. The approach is validated across 60 classifiers, including 48 ImageNet models, demonstrating that flipping fewer than ten bits can reduce accuracy to random guessing.

**Audience:**

Yes

**Audience Explanation:**

This work is highly relevant to the machine learning community, particularly for researchers and practitioners focused on AI safety, hardware security, and edge deployment. As deep learning models are increasingly deployed on mobile and IoT devices where physical access or hardware faults (e.g., Rowhammer attacks) are realistic threats, understanding the extreme fragility of model parameters without the need for training data is a significant insight.

**Claims And Evidence:**

Yes

**Claims Explanation:**

- The authors conduct extensive ablation studies, meticulously analyzing the impact of different scoring functions (comparing magnitude, gradient-based, and hybrid scores), layer selection strategies (early vs. late layers), and the specific efficacy of sign-bit versus exponent-bit flips.

- The experimental scope is impressive, covering 48 distinct ImageNet architectures (including ResNets, EfficientNets, and ViTs) as well as additional datasets like CIFAR and fine-grained classification tasks.

- Crucially, the authors extend their evaluation beyond standard FP32 models to include quantized (INT8) and binarized models, providing robust evidence that this vulnerability is fundamental to the parameter representation and not easily mitigated by standard compression techniques.

**Requested Changes:**

- **Clarification on the Novelty of the Hybrid Importance Scoring Function**: While the proposed "1-Pass" attack is effective, the underlying Hybrid Importance Scoring Function appears to be a linear combination of well-established pruning metrics—specifically Magnitude Pruning and Taylor Expansion/Hessian-based methods (like OBD/OBS). The authors should explicitly acknowledge the origins of these components in the main text. The review would benefit from a discussion clarifying that the novelty lies in the application of this combination to adversarial attacks rather than the derivation of a new mathematical metric.

- **Theoretical Justification for Heuristics**: A significant portion of the method relies on empirical heuristics, such as the "One-Flip-Per-Kernel Constraint" and the specific focus on sign bits. While the paper provides visual evidence (e.g., Figure 3) that a second flip might offset the damage, it lacks a deeper theoretical explanation. Why does this cancellation effect occur consistently across different architectures? Providing a more formal analysis or a broader statistical validation beyond visual examples would strengthen the paper.

- **Evaluation on Broader Tasks**: The current evaluation is heavily concentrated on image classification. To claim a generalized vulnerability in DNNs, it would be beneficial to assess if this fragility holds for other common computer vision tasks, such as Object Detection or Semantic Segmentation. These tasks have different loss landscapes and output structures; verifying if the "early layer lesion" hypothesis holds there would significantly increase the paper's impact.

- **Expansion on Defense Mechanisms**: The paper proposes a defense based on protecting the top-k parameters identified by DNL. It would be valuable to expand this into a more formal discussion on defense design. Specifically, is it possible to incorporate this vulnerability into the training process (e.g., via regularization or adversarial training against bit-flips) to make the weights naturally less sensitive, rather than relying solely on post-hoc hardware protection (ECC)? A brief discussion on "robust training" against DNL would be a valuable addition.

---

> ### Author Response · Authors · 2026-01-05
>
> Thank you for your thorough feedback,
>
> >*”Hybrid Importance Scoring Function appears to be a linear combination of well-established pruning metrics—specifically Magnitude Pruning and Taylor Expansion/Hessian-based methods (like OBD/OBS). The authors should explicitly acknowledge the origins of these components in the main text. The review would benefit from a discussion clarifying that the novelty lies in the application of this combination to adversarial attacks rather than the derivation of a new mathematical metric.”*
>
>
> We agree. In the revision, we will explicitly acknowledge that the 1P-DNL hybrid score is inspired by pruning saliency metrics: a magnitude term and a Taylor/OBD-style second-order term. We also appreciate your advice to clarify that our novelty is not in proposing a new pruning metric, but in using this combination for an adversarial application. We intend to add the following in the revision and would love to hear your thoughts:
>
> Our hybrid importance score used in 1P-DNL is inspired by classical pruning saliency criteria. The magnitude component corresponds to magnitude-based pruning, while the gradient/Hessian-based component follows Taylor-style saliency as in OBD/OBS under standard approximations. We do not claim a new saliency metric. Rather, our contribution is to adapt and evaluate this combination for adversarial parameter manipulation under a strict budget (no training data and at most a single forward/backward pass), where it consistently strengthens sign-bit flip attacks compared to using either component alone.
>
>
> >One-Flip-Per-Kernel Constraint, *”a second flip might offset the damage, it lacks a deeper theoretical explanation. Why does this cancellation effect occur consistently across different architectures? Providing a more formal analysis or a broader statistical validation beyond visual examples would strengthen the paper.”*
>
> Thank you for raising this point. Our “one-flip-per-kernel” constraint is motivated by an interaction that can arise when multiple sign flips occur within the same convolution kernel. A conv kernel computes a local linear response $y = w^\top x$ over a patch $x$. Flipping the sign of a single weight $w_i$ changes the pre-activation by $\Delta y = (-w_i - w_i)x_i = -2 w_i x_i$. With two flips inside the same kernel, $\Delta y = -2(w_i x_i + w_j x_j)$. Because natural-image patches have strong local correlations (neighboring pixels tend to have positive covariance), and because many early kernels resemble Sobel or Gabor-like edge detectors [1] with mixed positive/negative “lobes”, it is common for large-magnitude coefficients within the same kernel to have opposite signs. In that case, the two contributions $w_i x_i$ and $w_j x_j$ can partially cancel, so the second flip may reduce the net disturbance rather than compounding it. This is consistent with our observation that multiple flips in the same kernel often change orientation or slightly modify functionality rather than fully destroying the feature (Figure 3), motivating the constraint.
>
>
>
>
> To make the “cancellation” intuition concrete, we consider the typical magnitude of the pre-activation change using the mean-squared perturbation $\mathbb{E}[(\Delta y)^2]$ (the second moment). Unlike the mean change $\mathbb{E}[\Delta y]$, which may be close to zero due to sign cancellations over inputs even when perturbations are large, the mean-square captures the size (“energy/variance”) of the response. This choice also yields a closed-form dependence on the patch covariance, making the within-kernel cancellation mechanism explicit. For two flips in the same kernel,
>
> $\mathbb{E}[(\Delta y)^2] = 4(w_i^2 \Sigma_{ii} + w_j^2 \Sigma_{jj} + 2 w_i w_j \Sigma_{ij}),$
>
> where $\Sigma_{ij}=\mathbb{E}[x_i x_j]$. When $\Sigma_{ij}>0$ (correlated patch entries) and $w_i w_j<0$ (opposite-signed lobes), the cross-term is negative, reducing the net disturbance -i.e., a cancellation effect. Spreading flips across different kernels reduces the chance of such within-filter cancellation and yields more consistent damage per flip, which is exactly what the “one per kernel” constraint is designed to enforce. We will add this analysis to our revision.
>
> It is also worth noting that we provide concrete examples in Appendix B where a second flip in the same kernel partially offsets damage, while spending that flip in a different kernel compounds damage (e.g., MobileNetV3-Large and RegNet-Y 16GF).

---

> > ### Author Response · Authors · 2026-01-05
> > **Additional comment**
> >
> > >*”To claim a generalized vulnerability in DNNs, it would be beneficial to assess if this fragility holds for other common computer vision tasks, such as Object Detection or Semantic Segmentation.”*
> >
> > We agree, and we extended the evaluation beyond vision classification.
> >
> > **Object detection + instance segmentation (COCO2017).** We evaluated our standard DNL attack on:
> > 1) Mask R-CNN (FPN) with ResNet-50 and ResNet-101 backbones (torchvision weights)
> > 2) YOLOv8-seg (Ultralytics)
> >
> > We report AP@[0.50:0.95] and AP@0.50 for boxes (bbox) and masks (segm). Even k=1–2 sign flips are sufficient to collapse performance.
> >
> > | Model \ Backbone | Metric | Baseline | Post-attack (K=1) | AR(1) | Post-attack (K=2) | AR(2) |
> > |---|---|---:|---:|---:|---:|---:|
> > | ResNet-50 | bbox_AP | 0.38 | 0.01 | 97.36 | 0 | 100 |
> > |  | bbox_AP50 | 0.59 | 0.03 | 94.93 | 0 | 100 |
> > |  | segm_AP | 0.35 | 0 | 100 | 0 | 100 |
> > |  | segm_AP50 | 0.56 | 0.01 | 98.21 | 0 | 100 |
> > | ResNet-101 | bbox_AP | 0.4 | 0.01 | 97.51 | 0.01 | 97.51 |
> > |  | bbox_AP50 | 0.61 | 0.03 | 95.12 | 0.02 | 96.75 |
> > |  | segm_AP | 0.36 | 0 | 100 | 0 | 100 |
> > |  | segm_AP50 | 0.58 | 0.01 | 98.28 | 0 | 100 |
> > | YOLO-V8-SEG | bbox_AP | 0.33 | 0.05 | 83.66 | 0.05 | 86.33 |
> > |  | bbox_AP50 | 0.47 | 0.08 | 82.01 | 0.07 | 84.92 |
> > |  | segm_AP | 0.05 | 0.01 | 77.8 | 0.01 | 80.51 |
> > |  | segm_AP50 | 0.16 | 0.04 | 77.73 | 0.03 | 81.28 |
> >
> > **NLP (transformers).** We also ran experiments in the NLP domain with BERT, RoBERTa, and DistilBERT. Interestingly, unlike the vision models we tested, the most vulnerable bit we observed in these NLP models is often in the exponent.
> > Below we report AR(10) for our standard DNL attack on each model across three tasks: SST2, QNLI, and MRPC.
> >
> > | Model | Task | &nbsp;&nbsp;&nbsp; | AR(10) |
> > |---|---|:--:|---:|
> > | RoBERTa-base | sst2 | &nbsp; | 63.94 |
> > |  | qnli | &nbsp; | 49.99 |
> > |  | mrpc | &nbsp; | 65.32 |
> > | BERT-base | sst2 | &nbsp; | 50.11 |
> > |  | qnli | &nbsp; | 49.03 |
> > |  | mrpc | &nbsp; | 63.97 |
> > | DistilBERT-base | sst2 | &nbsp; | 49.04 |
> > |  | qnli | &nbsp; | 46.49 |
> > |  | mrpc | &nbsp; | 62.72 |
> >
> > We will include these additional results in the revision.
> >
> > >*”is it possible to incorporate this vulnerability into the training process (e.g., via regularization or adversarial training against bit-flips) to make the weights naturally less sensitive, rather than relying solely on post-hoc hardware protection (ECC)? A brief discussion on "robust training" against DNL would be a valuable addition.”*
> >
> >
> >
> > This is an excellent question, and one we have been asking ourselves as well.
> > We tried several training-time directions in a preliminary and somewhat exploratory way, but have not yet found a training regime that improves robustness to DNL. For example, we did not observe that common regularization already used in standard training pipelines (e.g., dropout) makes models noticeably more resilient to a small number of sign-bit flips.
> >
> > We also explored a more architectural intuition: limiting the flow between information pathways so that if a sign flip corrupts one pathway, the damage would be “contained” rather than propagating everywhere. In other words, we hoped to reduce cross-talk so most channels would not strongly depend on one another. Later we realized ShuffleNetV2 [2] already follows a related idea (illustrated clearly in Figure 4 of that paper), but in our experiments it was not more resilient than other architectures of comparable size. Our current hypothesis is that even with reduced cross-talk, there remain enough critical pathways that corrupting a single parameter can still cause collapse.
> >
> > Because of these observations, we chose to focus this paper’s defenses on post-hoc protection mechanisms. At the same time, we agree that “robust training” against bit flips is a valuable and open direction, and we explicitly invite future work to study it further in our “concluding remarks” section.
> >
> >
> > References:
> >
> > [1] How transferable are features in deep neural networks?
> >
> > [2] ShuffleNet V2: Practical Guidelines for Efficient CNN Architecture Design

---

> > > ### Author Response · Authors · 2026-01-16
> > > **Experiments on LLMs**
> > >
> > > Dear CEZm,
> > >
> > > Since our last response, continuing our discussion on NLP models, we also evaluated DNL and 1P-DNL on several LLMs. Remarkably, we found that k=1 (a single bit flip) is enough to completely disrupt generation: the model begins producing **nonsensical output and loses basic linguistic structure**.
> > >
> > > To make this quantitative, we tested three models: Qwen3-4B-Thinking, Qwen3-30B-A3B-Thinking, and Llama-3.1-Nemotron-Nano-8B-v1, and evaluated them on a stratified sample from the MATH500 benchmark (see [1] for technical details).
> > >
> > > All three models collapsed to 0% accuracy at k=1, as shown below:
> > >
> > > | Model | Baseline accuracy | Accuracy at DNL (k=1) | Accuracy at 1P-DNL (k=1) |
> > > |---|---:|---:|---:|
> > > | Qwen3-30B-A3B-Thinking | 98% | 0% | 0% |
> > > | Qwen3-4B-Thinking | 100% | 0% | 0% |
> > > | Llama-3.1-Nemotron-Nano-8B-v1 | 94% | 0% | 0% |
> > >
> > > Given the loss of coherent generation, the drop to 0% is unsurprising and would most likely repeat for any other benchmark.
> > >
> > > Importantly, in these LLMs the single-bit failure is not confined to a specific module. For example, in Llama-3.1-Nemotron-Nano-8B-v1 the flipped bit landed in an attention layer, but even when we constrain the search to FFN weights only, we still observe the same collapse to 0% at k=1.
> > >
> > > Notably, Qwen3-30B-A3B-Thinking is a mixture-of-experts model, and the flipped bit belonged to a single expert. Even though that expert is not active at every token, the generation was out of structure throughout the entire sequence. Our working hypothesis is that this is a direct consequence of autoregressive causality: it can be enough for the perturbation to corrupt part of the prompt’s latent representations, after which later tokens attend to these corrupted states and the error propagates and compounds across the context. This compounding effect is largely absent in standard classifiers (such as the image classifiers we reported in the paper), where each inference is independent, which may help explain why k=1 can be so destructive in LLM generation.
> > >
> > > [1] We evaluated using the EleutherAI lm-evaluation-harness (https://github.com/EleutherAI/lm-evaluation-harness), using "minerva_math" with its SymPy-based equivalence check (rather than strict exact match). We used a stratified sample by difficulty level (5 questions per level; 50 total) from MATH500.

---

### Decision · Action_Editor_kQHD · 2026-02-08

**Recommendation:** Accept as is

**Additional Comments:**

The revisions have effectively addressed reviewer concerns by clarifying the hybrid scoring function's novelty, providing theoretical backing for heuristics, and broadening evaluations to include object detection, NLP, and LLMs. The paper's empirical focus is a strength, demonstrating generalizability across models, but deeper theoretical analysis or expanded defensive strategies could be explored in future work. In the final version, please incorporate all reviewer-suggested modifications, such as further discussion of defense limitations and any remaining clarifications on methodology. The Featured Certification is recommended due to the work's significant novelty in revealing a simple yet impactful vulnerability.

**Audience:**

Yes

**Audience Explanation:**

The paper uncovers a novel vulnerability in DNNs at the parameter bit level, showing how simple sign bit flips can disrupt model performance via lightweight, data-free methods. This is highly relevant to the machine learning robustness, security, and adversarial attack communities within TMLR's audience, as it highlights practical implications for software, firmware, and hardware exploits, while also proposing partial defenses. The findings could influence future research on model security and attract interest from practitioners concerned with real-world DNN deployments.

**Claims And Evidence:**

Yes

**Claims Explanation:**

The claims are well-supported by extensive empirical evidence, including experiments across multiple DNN architectures, tasks, and datasets. The authors demonstrate catastrophic accuracy drops through clear, reproducible results. Revisions have added theoretical justifications for heuristics and addressed novelty concerns, enhancing rigor. While the work is primarily empirical, the evidence is convincing and transparently presented, with limitations on defenses acknowledged.